



# A data derived workflow for reservoir operations for simulating reservoir operations in a global hydrologic model

Jennie C. Steyaert[1], Edwin Sutanudjaja[1], Marc Bierkens[1,2], and Niko Wanders[1]

[1]Utrecht University, Department of Physical Geography, Princetonlaan 8a, Utrecht, The Netherlands
[2]Deltares, Unit D Subsurface and Groundwater Systems, Utrecht, The Netherlands

**Correspondence:** Jennie C. Steyaert (j.c.steyaert@uu.nl)

**Abstract.** Globally there are over 24,000 storage structures (e.g. dams and reservoirs) that contribute over 7,000 $km^3$ of storage. Until recently, most of the data regarding these reservoirs was not openly accessible. As a result, many studies rely on generalized operations based on generalized assumptions about reservoir storage dynamics and management. With the creation of global datasets such as the Global Reservoirs and Dams (GRanD), RealSat, GloLakes, and the International Coalition for

Large Dams database (iCOLD) as well as localized datasets such as ResOpsUS for the contiguous United States, and the Mekong Data Monitor for the Mekong River basin, the inference of reservoir operations using data derived techniques has become much more ubiquitous regionally. Yet to our knowledge, there has been no global application of data-derived methods due to their model complexities and data limitations. Therefore, our analysis aims to fill this gap by providing a workflow for implementing data derived reservoir operations in the large scale hydrologic models with an application in the PCR-

GLOBWB 2 global hydrologic model. This methodology uses global satellite altimetry data from GloLakes, a parameterization methodology developed by Turner et al. (2021), and a random forest model. We also test the sensitivity of our reservoir scheme to downstream demand by selecting three different downstream areas presumed to be served by reservoirs release (command areas): 250 km, 650 km, and 1100 $km$. Our results demonstrate that our random forest algorithm is able to capture the storage dynamics and that the errors are mainly due to the errors in using remotely sensed storage data. Additionally, we observe in

many cases that deriving operational bounds from historical reservoir time series has minimal impact on streamflow at the basin outlets nor is the scheme sensitive to the downstream command areas. We do observe that streamflow is affected directly downstream from the reservoirs and that the data-derived methodology does increase the accuracy of simulated global reservoir storage when compared to observations. In fact, the derived operations have much lower storage values that align better with both direct and remotely sensed reservoir storage observations. This demonstrates that generic operations overestimate the total

amount of water stored in reservoirs and, as a result, are potentially overestimating water availability. Ultimately, our workflow allows global hydrologic models to capitalize on recent data acquisition by remote sensing to provide more accurate reservoir storage and global water security.



# 1 Introduction

Across the globe, there are over 24,000 reservoirs that regulate 63% of all global rivers (Hou et al., 2024), contain 61% of the
global seasonal variability in water storage (Cooley et al., 2021), and greatly decrease river connectivity (Grill et al., 2019;
Belletti et al., 2020). With this loss of river connectivity comes a large amount of water storage (over 8,000,000 $m^3$ (Lehner
et al., 2011)) that provides water for a variety of main purposes ranging from water supply and irrigation in more arid climates
to hydropower and flood control in more humid environments. For many reservoirs, these uses are not unique as one reservoir
may provide multiple uses integral to water management regardless of their primary use (Belletti et al., 2020; Biemans et al.,
2011; Boulange et al., 2021; Doll et al., 2009; Grill et al., 2019). With projected increases in global cropland (up to 1,244 $MHa$)
and over 90% of the world's population living within 10 $km$ of readily available surface water (Kummu et al., 2011; Potapov
et al., 2021), the importance of reservoirs for flood control, irrigation, and water supply is large as the majority of society is
dependent on reservoirs for a variety of uses (Di Baldassarre et al., 2018). Therefore having an accurate depiction of reservoir
storage at the global scale in observations and models is crucial to evaluating both historical and projected water availability.

Reservoir operating policies strongly impact the seasonality and variability in downstream streamflow regimes, yet their op-
erations in global hydrologic models are highly dependent on data availability. Across the globe, reservoirs decrease streamflow
variability with heightened effects during extreme flows (Salwey et al., 2023; Zajac et al., 2017; Chalise et al., 2021). However,
capturing this behavior in hydrological modeling tools is far from trivial due to a lack of observational data that describes
reservoir properties and operational policies. In many generic operations, streamflow sensitivity to reservoirs in large-scale
hydrologic models is quite limited with only a few regions experiencing large changes (Biemans et al., 2011; Hanasaki et al.,
2006; Haddeland et al., 2006) Therefore observational data is required to support the development of parameterizations that
can replicate reservoir dynamics and be used to assess downstream impacts. Ultimately, the analysis of these impacts on the
global scale is now largely limited by the uncertainty in the parameterization of reservoir operation in hydrologic models which
leads to varied effects on the modeled streamflow regimes (Zajac et al., 2017). Therefore, more observational reservoir data is
needed to remedy the current biases and uncertainty within reservoir operations.

Even with the current data limitations, global hydrologic models can use generic operating curves to approximate reservoir
releases and analyze the impact of water demands on global reservoir storage. In general, global reservoir storage capacity
increased over the past 40 years as dams were built to support a variety of main uses (Lehner et al., 2011; Haddeland et al.,
2006; Li et al., 2020; Wisser et al., 2013). This increase in storage capacity does not necessarily correlate with an increase
in storage as many regions (primarily arid basins such as southeastern Australia, southwestern US, and eastern Brazil) have
observed decreases in total reservoir storage (Steyaert and Condon, 2024; Li et al., 2020) due in part to increased domestic
water demand resulting from population growth and decreased storage capacities from sedimentation (Simeone et al., 2024;
Li et al., 2020; Wang et al., 2024; Wisser et al., 2013). This regionality is especially important as remote sensing observations
show that most global reservoirs (more than 50%) have not filled between 2010-2022 and many reservoirs in the Southern
Hemisphere observe strong declines (Yao et al., 2023; Wang et al., 2024; Li et al., 2023). Ultimately, the regional differences in
reservoir storage, the lack of observed filling, and the impact of sedimentation and water demand on storage levels are not well



captured in generic operations due to simplified calculations that do not assimilate information derived from observed storage values.

The first generation of generic reservoir operations by Meigh et al. (1999), Döll et al. (2003), Pietroniro et al. (2007), and Rost et al. (2008) used static reservoir capacities and a water balance method to develop simplified generic operating curves. In these methodologies, researchers need a total storage capacity (usually derived from static datasets such as GRanD (Lehner et al., 2011) or the World Registry of Dams from the International Commission of Large Dams (ICOLD; https://www.icold-cigb.org) and/or surface area. These two values are used to denote how much water could be stored in a given reservoir for a given time step and any "excess" water is released downstream. These methodologies are more readily incorporated into large-scale hydrologic models, such as WaterGAP (Doll et al., 2009) and PCR-GLOBWB 2 (Sutanudjaja et al., 2018) due to their limited data requirements. Their main critique lies in the assumption that reservoirs are usually filled to their maximum capacity which overestimates the amount of storage (Steyaert and Condon, 2024; Salwey et al., 2023). Additionally, the lack of different reservoir purposes may result in generic operations that differ greatly from the observed operational policies (Steyaert and Condon, 2024; Turner et al., 2021). More recent advances on these fronts by Salwey et al. (2023) and Brunner and Naveau (2023) focus on utilizing water balance methods to back-calculate transient reservoir characteristics from openly available data such as regional streamflow. These methodologies show promise regionally but have not yet been scaled to global applications nor implemented in global hydrologic models (Hosseini-Moghari and Döll, 2024).

The above generic operations have historically utilized static reservoir characteristics (e.g. location, capacity, area, main purpose) from datasets such as the Global Reservoirs and Dams dataset (GRanD Lehner et al., 2011) and World Registry of Dams from iCOLD. The majority of these datasets include data for the largest dams across the globe and provide valuable input as most hydrologic models are typically run on coarser spatial scales (0.5 degrees or around 50km at the equator). However, recent developments in hyper-resolution modeling have resulted in a higher demand for high-resolution information on reservoir properties, especially locations and operations (e.g. Hoch et al., 2022; van Jaarsveld et al., 2024). While datasets like GRanD and iCOLD contain the majority of storage across the globe ($8*10^6 \ m^3$), these two datasets do not include all storage structures nor do they contain transient reservoir characteristics that can be used to develop operational curves. In fact, GRanD, the most used reservoir dataset for evaluation of reservoir operations only contains 6,000 global structures with a storage of 6,197 $km^3$ as it is focused on dams greater than 0.1 $km^3$. More recent mapping by Wang et al. (2022) mapped 24,000 structures with a total storage of over 7,000 $km^3$; however, this new dataset still does not contain transient reservoir operation information. These static characteristics are useful for implementing generic reservoir operations to analyze the impact of reservoirs on the larger hydrologic cycle, however, the lack of transience and defined seasonality can limit accurate storage simulations and in many cases can cause models to overestimate the amount of reservoir storage (Steyaert and Condon, 2024; Brunner and Naveau, 2023; Salwey et al., 2023).

More complex models of reservoir operations pushed for the inclusion of water demand and main use as a driving factor for reservoir releases in order to better represent reservoir operations. Initially, the work by Hanasaki et al. (2006) accounted for reservoir releases at a yearly time step and included demand in the reservoir release equation. In addition to this, Hanasaki et al. (2006) also included different operational policies based on the main reservoir purpose and employed a release factor



(a ratio made up of current storage, maximum capacity, and dead storage) to denote how much extra water might need to be accounted for based on how full the reservoir is. Similar to this approach, Biemans et al. (2011), Haddeland et al. (2006), Voisin et al. (2013),and van Beek et al. (2011) also included downstream demand into the reservoir water balance as well as

incorporating different operational policies based on the reported main reservoir use. That said, only four main categories are typically employed: irrigation, flood control, hydropower generation, and navigation. In the case of van Beek et al. (2011) and Voisin et al. (2013) only irrigation and hydropower uses are used. While van Beek et al. (2011) changed operations based on main use, Voisin et al. (2013) changed operations based on seasonality (i.e. flood control operations in the spring and irrigation prioritization in the summer/autumn). Ultimately, these operational policies are also easy to incorporate into large-

scale hydrologic models as their calculations rely solely on static reservoir capacity, modeled inflow, modeled downstream demand, and one or two extra parameters that can either be readily calculated within the model or calibrated using additional observations.

   With the inclusion of water demand another factor of uncertainty is introduced: what is the downstream command area (i.e. the downstream area dependent on the upstream reservoir for water supply) that influences the reservoir release decisions?

While numerous studies have employed downstream demand in their reservoir operations (Yassin et al., 2019; Turner et al., 2021; Hanasaki et al., 2006; van Beek et al., 2011; Biemans et al., 2011; Voisin et al., 2013; Haddeland et al., 2006), only three main studies provide the size of these command areas. In all cases, the downstream area is defined as the stream reach downstream from the reservoir outlet to the next reservoir or the river mouth, whichever comes first. Haddeland et al. (2006) opted to use a downstream distance of 250 $km$ (or 5 grid cells at a spatial resolution of 0.5 degrees). They also employed a

check that ensured each dam had at least one downstream demand pixel and split the demand based on the ratio of reservoir capacities (meaning a larger reservoir would meet more demand than a smaller one). This value was updated by van Beek et al. (2011), who observed that the most operational downstream area would be 600 $km$ (or the distance a water droplet could travel in one week). Hanasaki et al. (2006) used a downstream command area of 1100 $km$ which is equivalent to the distance one water particle could travel in a month, which was the temporal resolution of the employed hydrologic model. The difference

in how far downstream a reservoir should operate allows operational policies to be more sensitive to the short-term variations (such as using a downstream area of 250 $km$ Haddeland et al., 2006) or to the longer-term changes in a system (600-1100 $km$ van Beek et al., 2011; Hanasaki et al., 2006). These differences are useful for analyses but aside from choosing a value based on model characteristics, the actual sensitivity of modeled reservoir releases to differences in command areas has not yet been studied.

Reservoir operation schemes are also adapting to the newly available data in both the regional and global context. Turner et al. (2021); Zhao et al. (2016); Burek et al. (2020); Macian-Sorribes and Pulido-Velazquez (2020), and Yassin et al. (2019) parameterized reservoirs based on available data using anywhere from 10 to 72 key parameters. The majority of these methods utilized reservoir data that was freely accessible via online platforms and opted to include operational bounds based on different operational decisions (a level for flood control, an active storage level, a conservation pool, and a dead storage zone). Burek

et al. (2020) and Zhao et al. (2016) pioneered this work by dividing reservoirs into multiple levels based on multiple linear regressions (Burek et al., 2020) and using user-derived parameters obtained from the local stakeholders (Zhao et al., 2016).





While these methodologies showed promise regionally, the lack of global data removes the individual operating nuances and can result in errors in downstream flows specifically in regions that are underrepresented (Yassin et al., 2019; Turner et al., 2021). Yassin et al. (2019) and Turner et al. (2021) furthered this methodology by using freely accessible historical reservoir time series to obtain operational curves based data mined reservoir data. This data was used to create seasonal time series of storage and release patterns that (depending on the number of variables) can be more easily incorporated into regional models (e.g. MOSART for Turner et al., 2021).

While the majority of these schemes analyze regional dynamics, only a few have been incorporated into regional models: Turner et al. (2021) into MOSART, Yassin et al. (2019) into the Canadian land surface model MESH (Pietroniro et al., 2007), and Macian-Sorribes and Pulido-Velazquez (2020) into RiuNet, with generally good results. The incorporation into large-scale hydrologic models has, until recently, been very limited by the high data requirements, reduced generalizability, the number of interconnected variables, and the number of parameters in these models. The lack of available data, for example, makes training and obtaining these parameters quite challenging as many global hydrologic models strive to quantify water quantity on the global scale instead of the local scale and thus consistent high-quality input data needs to exist with global coverage. Additionally, the incorporation of multiple parameters and inter-dependencies (i.e. large-scale hydrologic models require reservoir storage, yet reservoir storage calculations from data-driven models are dependent upon output from hydrologic models) which increases the computational load and the potential for compounding errors already associated within the original hydrologic model as well as the reservoir model.

The advancement and availability of high-resolution transient regional data has led to a rapid increase in the application of machine learning in reservoir operations, specifically focused on networks (Coerver et al., 2018) and fuzzy logic schemes (Macian-Sorribes and Pulido-Velazquez, 2020). While these methods have seen large improvements in accurately depicting reservoir storage and release, the amount of data required for their development, their complexity, and the numerous required parameters can make linkages with global hydrologic models challenging. Especially since the transferability of these locally parameterized models is limited and the high number of parameters makes it difficult to extrapolate these models to global-scale applications. Therefore, there have not yet been any global applications of these machine learning operational schemes.

Recent improvements in satellite altimetry data (Chen et al., 2022; Hou et al., 2024) as well as the plethora of public and private satellite missions over the past 20 years have made the creation of transient reservoir operation datasets (such as GloLakes Hou et al., 2024) and updated reservoir characteristics (GeoDAR Wang et al., 2022) possible. This increase in satellite monitoring allows for the creation of reservoir surface area time series that can be input to statistical models to back-calculate reservoir storage at select temporal and spatial resolutions. In general, surface areas are mapped via LandSat imagery using a combination of machine learning and manual approaches (Hou et al., 2024; Lehner et al., 2011; Ling et al., 2020; Zhang et al., 2019). After pixels are classified as reservoirs, the digital elevation model is used to invert the reservoir storage. Historically, remote sensing methodologies have used Sentinel or GREALM as their DEM (digital elevation model) component (Zhao et al., 2016), but increases in satellite missions and the increased need for monitoring glacier and ice changes led to a new satellite called IceSAT which has shown promise in monitoring lake elevations using Lidar measurements (Chen et al., 2022). Once the DEM, and surface area are obtained for each site, a statistical model is used to estimate the reservoir storage





at each measurement point over the entirety of the time series(Hou et al., 2024; Chen et al., 2022; Zhao et al., 2016). Using these datasets, Hou et al. developed a global dataset of historical reservoir storages that can be used to evaluate global reservoir storage fluctuations, streamflow impacts, and to update reservoir release models.

To facilitate accurate global hydrologic modeling at high spatial resolutions, we need to combine the simplicity and generalizability of classical reservoir modeling approaches with the newly available data and data-driven operating rules. Specifically, there is a need to better incorporate transient reservoir operations into global hydrologic models. Therefore, the objective of this study is to utilize global reservoir storage data calculated via satellite altimetry in combination with a data-driven framework to derive static and seasonal parameters for global reservoir operation. We will follow the work of Turner et al. (2021)

by estimating seasonal boundaries for release and conservation. Ultimately, these components will be placed into the global hydrologic model PCR-GLOBWB 2 (Sutanudjaja et al., 2018) to answer the following questions:

– Are reservoir operations whose operating rules have been extrapolated using data-derived approaches more accurate than generic ones and what does this mean for the current data gaps in reservoir operational data?

– What is the sensitivity of different reservoir operations to the size of the command area

– How does the ability of global hydrologic models to reproduce streamflow and reservoir storage improve or decrease depending on the type of reservoir operation used (i.e. generic vs data-driven)?

To answer these questions, this study uses a dataset of remotely sensed reservoir time series from Hou et al. (2024) that can be the basis for a statistical model (STARFIT) which derives reservoir operational bounds. Using these operational bounds, we derive two main reservoir models for *irrigation-like* and *hydropower-like* dams. We implement these operational policies in

PCR-GLOBWB 2 in order to evaluate the global impact of changing reservoir regimes on global water resources.

## 2 Methods

For this analysis, we updated the number of storage structures in PCR-GLOBWB 2 (Sutanudjaja et al., 2018) from 6,000 to over 24,000 worldwide using a new dataset: GeoDAR (Wang et al., 2022) (Section 2.1). Additionally, we also update the current reservoir operations. To do this, we followed the data derived workflow outlined in Figure 1. First, we utilized a new dataset of

remotely sensed reservoir surface area and estimated storage created by Hou et al. (2024) called GloLAKES (titled reservoir time series in Figure 1). We input this weekly data into the STARFIT model developed by Turner et al. (2021) (Section 2.4.2) to determine seasonal trends of flood and conservation zones from storage data. After obtaining flood and conservation curves for 1752 reservoirs, we then trained a random forest model to predict the ten parameters that determine the flood and conservation zones for these 1752 reservoirs. The random forest model used static reservoir characteristics, and hydroclimatic socioeconomic

variables as features. We then used the trained random forest model (Section 2.5) to extrapolate active storage bounds for all of the 24,000 structures and compared this new operational scheme with the current reservoir scheme in PCR-GLOBWB 2 (Section 2.6). Finally, we split the 22,000 dams into categories based on their main use and modelled releases based on two





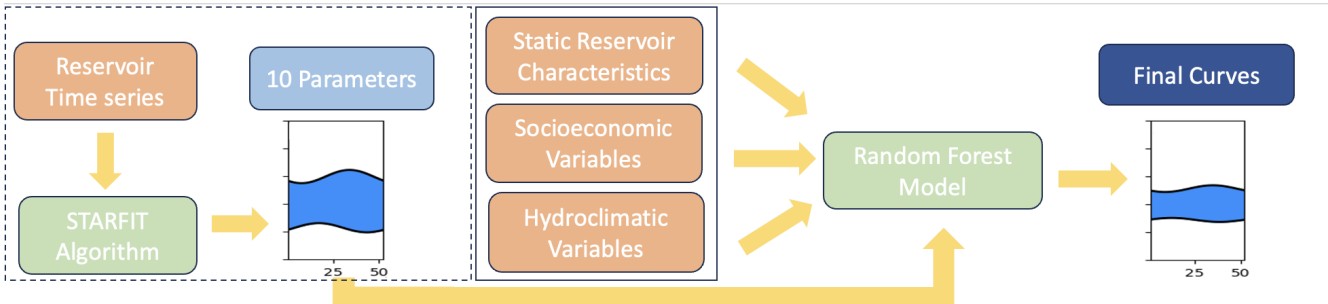

**Figure 1.** Outlines our workflow to create the data-derived operations implemented in PCR-GLOBWB 2. Orange boxes denote the instances where outside reservoir, socioeconomic, or hydroclimatic datasets were used. Blue boxes denote the parameters and the final curves, while green boxes denote the models that we used in our methodology. For our analysis, we used 1752 reservoir time series and thus the STARFIT maximum-minimum active zone boundaries are only available for 1752 dams. The Random forest methodology allows us to extrapolate the active zone curves from these 1752 structures to 24,000 global reservoirs.

main operational schemes described in Section 2.4.3 and Section 2.4.4. This allowed us to analyze the bias in using different input data in our random forest workflow and the impact of operational schemes on PCR-GLOBWB 2 outputs.

## 2.1 Data Sets

To compliment the hydrologic model and statistical model (STARFIT from Turner et al. (2021)), we required a variety of datasets that are depicted in Table 1. These datasets are used to update the number of reservoirs in PCR-GLOBWB 2, train the random forest algorithm, and validate our analysis.



| Dataset Name | Citation | Type of Data | Purpose |
|---|---|---|---|
| Global Runoff Data Centre | https://www.bafg.de/ GRDC/EN/Home/ homepage_node.html | Discharge data | Streamflow Validation |
| GeoDAR | Wang et al. (2022) | Static reservoir characteristics | Base Maps |
| Global Reservoir and Dams Dataset | Lehner et al. (2011) | Static reservoir characteristics | Base Maps |
| International Commission of Large Dams Dataset | https://www.icold-cigb. org | Static reservoir characteristics | Gap filling missing reservoir purposes and construction dates |
| GloLAKES | Hou et al. (2024) | Transient reservoir operations | Random forest algorithm and storage validation |
| ResOpsUS | Steyaert et al. (2022) | Transient reservoir operations | Storage validation |

**Table 1.** Describes the data used in this analysis. Column one gives the dataset name. Column two gives the paper citation or webpage associated with the data source. Column three gives the type of data and column four describes the function the data played in this analysis

### 2.1.1 Reservoir Data

To update the number of reservoirs in PCR-GLOBWB 2, we used a newly minted reservoir dataset that contains over 24,000 global structures (Wang et al., 2022). The dataset, itself, contains static reservoir properties such as surface area, capacity, latitude, and longitude. In order to fill missing gaps in surface area, capacity, creation date, and main reservoir purpose, we linked GeoDAR (Wang et al., 2022) to the Global Reservoirs and Dams Dataset (GRanD) created by Lehner et al. (2011) and the International Commission of Large Dams (iCOLD) registry (ICOLD; https://www.icold-cigb.org, last access: 13 March

2019). From this updated table, we created yearly maps of static reservoir characteristics which are used to model reservoir releases and to distinguish between two operational policies *hydropower-like* and *irrigation-like*.

In addition to static reservoir characteristics, we required transient reservoir data for the STARFIT model. For this, we used the GloLAKES dataset (Hou et al., 2024), a dataset that contains remotely sensed reservoir surface area time series data for the majority of dams and lakes in HydroSHEDS (Giachetta and Willett, 2018). Hou et al. (2024) used remotely sensed surface

area extents from LandSat and reservoir depths from BLUEDOT/Sentinel-2. These outputs were then put into a storage area relationship from Crétaux et al. (2016) to calculate reservoir storage. From GloLakes, we fit curves for over 6000 storage structures worldwide. Due to limited satellite coverage and temporal gaps, the majority of data points sit within latitudes between 56° South and 82.8° North and are only available on the weekly time scale.





## 2.2 Validation Data

To validate our analysis, we used two types of validation data to benchmark the quality of the random forest model and our data-driven reservoir operations: storage data from ResOpsUS and the LandSAT+ICESAT product from GloLAKES, and streamflow data from the Global Runoff Data Center. Reservoir storage data from the ResOpsUS dataset Steyaert et al., 2022, which contains over 600 reservoirs spread throughout the conterminous United States and was the original data Turner et al. used to create the STARFIT algorithm, is used as an independent benchmark to evaluate how well the workflow captures

seasonalities against direct observations. To validate storage changes and potential storage uncertainties within the workflow globally, we used the estimated water level variations from LANDSAT+Sentinel2 and LandSAT+ICESAT as both are available at the same temporal and spatial resolution. To validate and benchmark the addition of our data-driven operations in PCR-GLOBWB 2, we used global streamflow data from over 10,000 gages in the Global Runoff Data Center's (GRDC) dataset. The data in GRDC has good coverage over North America and Europe and lower coverage in Asia and Africa (Burek and Smilovic,

2023). Even with the limited coverage, GRDC data is still the best for global validation of streamflow. Our validation used 75% of all the gages and of these 75%, 8% of the gauges have a full period of record from 1979-2023 which is the period over which we analyzed changes in streamflow due to reservoir operations. For a more detailed analysis, we removed all the gages that were not directly downstream of a reservoir and were therefore left with 2,666.

All the data for this analysis was acquired through open-access sources and has been noted in the Data Availability section.

## 230 2.3 PCR-GLOBWB 2

PCR-GLOBWB 2, our primary global hydrologic model, is a grid-based hydrologic model that covers the entirety of the globe (aside from Greenland and Antarctica) and estimates human water use as well as hydrological variables. The computational grid of PCR-GLOBWB 2 is available at a variety of resolutions: 30 arc minutes, 5 arc minutes (Sutanudjaja et al., 2018) or 30 arc seconds (van Jaarsveld et al., 2024; Hoch et al., 2022). For each grid and time step (daily for hydrologic variables and

dynamic for river routing), the model simulates moisture storage and water exchanges between the ground, atmosphere, and soils. Through this, it can effectively simulate transpiration from crops and vegetation, evaporation from soil and open water, snow and glacier processes such as accumulation and melt, surface runoff, groundwater recharge, soil and plant transpiration, discharge, reservoir storage, reservoir release, and runoff. The system's runoff is routed through a river network to potential sinks such as the ocean, or endoheric lakes and wetlands using the kinematic wave approach. The model currently has 6,000

dams based on GRanD (Lehner et al., 2011) that use a generic operation scheme developed by Sutanudjaja et al. and described in Section 2.4.1. In addition to modeling hydrologic variables, Wada et al. (2014) included an updated scheme for evaluating human water use. At each daily time step in PCR-GLOBWB 2, three main steps occur: 1) water demand for irrigation, industrial, livestock, and domestic uses is estimated, 2) these estimated demands are translated into withdrawals from surface and/or groundwater sources subject to the availability of these resources and the maximum groundwater pumping capacities, and 3)

consumptive water use and return flows are calculated per sector (Sutanudjaja et al., 2018).





### 2.3.1 Inclusion of GeoDAR into PCR-GLOBWB 2 domain

Since PCR-GLOBWB 2 runs on a gridded model domain, our first step in updating the number of structures was to create new input maps by remapping the geospatial structures from GeoDAR to the PCR-GLOBWB 2 domain. First, we rasterized the GeoDAR attributes and overlaid them on the PCRaster global domain maps that has a 5 arc minute spatial resolution (about 10 km at the equator). This spatial resolution is the optimal balance between computational demand and model performance and has been extensively validated and benchmarked (Sutanudjaja et al., 2018). Due to the decision to model at a spatial resolution of 5 arc minutes, we found that some dams are situated within the same grid cell. For these grid cells that had multiple dams, we summed their storage capacity and chose the construction date of the first dam. After remapping the dams to their new location on the PCR-GLOBWB 2 domain, we then ensured that reservoirs were not split between multiple catchments and that lake and reservoirs were not mixed if situated in the same grid cell. Here, we calculated the total area of each lake and reservoir per cell and if the total area of the lake was larger than the reservoir for overlapping structures, we converted the reservoir cells to lakes and visa versa. We ensured that reservoirs were only present in the dynamic model simulation after their initial construction date to avoid unrealistic discharge simulations. Lastly, we repeated this process with the original input data from GRanD to ensure any differences in our results could be solely attributed to the reservoir operations or initial reservoir input data.

### 2.4 Reservoir Operations

To evaluate the impact of "standard" generic operations and data-driven reservoir operations on global discharge, we opted to utilize two different reservoir schemes. The first is a generic reservoir scheme derived from Sutanudjaja et al. (2018) which uses static values such as maximum reservoir capacity and operational bounds: 10% of maximum storage capacity as the dead storage (the point where water can no longer be abstracted from the dam) and 75% of maximum storage capacity as the maximum available storage, with year-round fixed boundaries that do not reflect seasonality. The second is a data-derived reservoir scheme: STARFIT, derived by Turner et al. (2021) to fit historical reservoir time series from ResOpsUS (Steyaert et al., 2022). This method derives weekly operational bounds for flood and conservation zones from historical storage time series and has been calibrated and tested in the United States (Turner et al., 2021). As our analysis is all done globally, we use data from 1752 dams in GloLAKES (Hou et al., 2024) and derive the operational bounds for the STARFIT using a combination of observations and machine learning. From there, we employ the two different operational schemes based on the main purpose: Section 2.4.4 and Section 2.4.3)

For all of the operational schemes (generic, *irrigation-like* and *hydropower-like*, we employ the following method. First, we define the initial release and set up the framework for other conditions such as environmental flows, and floods. To do this, we first calculate the initial discharge into the model that is defined by equation 1.

$$R_i = RF * R_{avg} \tag{1}$$

where RF is the reduction factor as shown in equation 3 with the updated values as defined above, $R_{avg}$ is the long-term average outflow that is dynamically calculated within the model. This initial release ($R_i$) is used as a starting point





in each operational scheme and is modified based on the type of operation used. In all cases, the $R_i$ cannot be greater than the
downstream bankfull discharge value of 2.3 which is the highest ratio of streamflow to stream network is expected to transport
without floods occurring.

    Then, we calculate the new storage using the following water balance equation under the initial release to ensure a starting
point of storage:

$$S_c(t+1) = Sc(t) + I(t) + P(t) - E(t) - R_i \tag{2}$$

where I is inflow, A is area reservoir, P is precipitation, E is evaporation, R is release.

    We use this updated storage to define the reduction factor, which is the fraction of the long-term average reservoir release,
as outlined in equation 3.

$$RF = \frac{S_c - S_{min}}{S_{max} - S_{min}} \tag{3}$$

    where $S_c$ is the current storage level, $S_{max}$ is the maximum storage capacity and $S_{min}$ is the dead storage level.

After this, we employ updated release calculations based on Section 2.4.1, Section 2.4.3, or Section 2.4.4 depending on the
type of operational scheme used as well as the type of dam. This new release is then placed back into equation 2 to update the
storage at the current timestep.

### 2.4.1 Generic Reservoir Scheme in PCR-GLOBWB 2

Generic reservoir operations are already implemented in PCR-GLOBWB 2 by Sutanudjaja et al. (2018) and mimic that of
hydropower operations. Each reservoir has an active zone between 10% and 75% of the reported maximum storage capacity
in GRanD. When storage is below the dead storage limit (i.e. 10% of maximum capacity), the reservoir does not release any
water. Between 10% and 75% full, the reservoir outflow is scaled by the reduction factor denoted by equation 3

    The generic release operations are defined by the following piecewise function.

$$R = \begin{cases} 0 & \text{if } S_c < S_{min} \\ RF * Q_{avg} & \text{if } S_c > S_{min} \text{ and } S_c < S_{max} \\ \frac{S_c - S_{cap}}{S_{max} - S_{cap}} * (Q_{bf} - Q_{avg}) + B & \text{if } S_c > S_{max} \end{cases} \tag{4}$$

where $Q_{avg}$ is the longterm average discharge at the point location of the dam, $Q_{bf}$ is the bankfull discharge, and B is the
bankfull number which is the ratio of bankfull discharge to the average discharge and is denoted as 2.3in our analysis (van
Beek et al., 2011).

    In all instances, demand is set at zero meaning that the reservoir is changing releases solely based on storage. When reservoir
storage minus the projected release would be greater than the set maximum of 75% of storage capacity or $S_{max}$, the reservoir
enters flood conditions. At this point, an additional release is added to ensure that the reservoir is brought back to the upper
75% of storage and is not over-topped. All the water is routed downstream where it can be further allocated within the river
network in PCR-GLOBWB 2.





### 2.4.2 Data Driven Reservoir Operations - STARFIT

To remedy gaps in generic reservoir operations (primarily the lack of demand, no environmental flow, and uniformly large

active zones), Turner et al. (2021) used the ResOpsUS dataset (Steyaert et al., 2022) to create STARFIT, a reservoir model that takes historical reservoir storage, release, and inflow data to output operational ranges for each variable. To do this, these daily storage, release and inflow values are aggregated into weekly time series and a combination of sine and cosine curves (described by equation **??** below) are fit to the upper and lower percentiles of each time series. This results in two curves bounded by the upper and lower percentiles with a total of 10 parameters which creates a range within which reservoir release,

storage, and inflow should sit based on historical data(Turner et al. (2021) used the period from 1980 - 2020).

$$S_t = \mu + \alpha * sin(2\pi\omega t) + \beta * cos(2\pi\omega t) \tag{5}$$

For our analysis, we were only able to obtain storage time series from GloLakes, as no global dataset of reservoirs releases is available. We applied the Turner algorithm to GloLAKES storage data for 1752 dams from 1980 - 2020. This data is already aggregated weekly due to the gaps in satellite passages and therefore we were able to use the estimated storage values directly

from these remotely-sensed observations. We did not gap-fill the data as this would add additional assumptions that could later lead to increased uncertainties. We fit flood and conservations curves to the 1752 dams and obtained the 10 parameters (five for the flood and five for the conservation curve) denoting the upper and lower limits of the active zone. The 10 parameters were then used as training data for a random forest model that was used to extrapolate the parameters to the global scale (Section 2.5).

Since there are large differences between irrigation/water supply dams and hydropower/navigation, we grouped the dams into two main categories. Main purposes such as irrigation and water supply were grouped as their dynamics are driven by downstream demand (either for agriculture or for domestic uses), while main purposes that have operations that are not demand-driven such as hydropower, fisheries, navigation, recreation, and other were grouped into a second category. This grouping allowed us to have two main operational strategies: one for *hydropower* type dams which strive to keep storage as

high as possible and one for *irrigation* type dams which strive to meet downstream demand.

Before we could implement the data-derived reservoir operations in PCR-GLOBWB 2, we had to calculate command areas as the reservoir operation scheme requires downstream demand to determine how much water should be released by water supply and irrigation dams. Based on our literature review, we observed that there are three main command areas used by large-scale hydrologic models: 250 $km$ (Haddeland et al., 2006), 600 $km$ (van Beek et al., 2011), 1100 $km$ (Hanasaki et al.,

2006). For each dam represented in the GeoDAR map, we calculated the farthest location along the river network at a distance of 250 $km$, 600 $km$, or 1100 $km$ and allocated the water demand in each cell up to this point to the upstream reservoir. If during this process, another dam intersects the river network before the full command area is created, we assume that this is the maximum distance that is served by the upstream reservoir.

Hydropower-like dams are only able to meet a portion of downstream demand when they are within their active zone

and meeting demand would not decrease storage below the conservation bounds. Irrigation-like dams, on the other hand, are



allowed to meet demand when in the active zone and above dead storage (denoted as 10% of the storage capacity). We also included surface water abstractions for irrigation-like dams.

Lastly, we implement a piecewise function for releases based on the current reservoir storage ($S_c$) where $R_f$ is the flood release and $R_i$ and $R_h$ are the irrigation and hydropower releases in the active zone and are described in by equation 9 in Section 2.4.4 and by equation 8 in Section 2.4.3 respectively.

$$R = \begin{cases} R_i + R_f, & \text{if } S_c - R \geq S_{cap} \\ R_h & \text{if } S_c > S_{min} \text{ and } S_c < S_{max} \text{ and use is hydropower} \\ R_i & \text{if } S_c > S_{min} \text{ and } S_c < S_{max} \text{ and use is irrigation} \\ Env & \text{if } S_c < S_{min} \text{and } R_i \text{ or } R_h < Env \text{ and } S_c - Env > 0 \end{cases} \quad \text{where } R_f = S_c - R - S_{cap} \quad (6)$$

### 2.4.3 Operations for Hydropower-like Dams

For the hydropower-like dams, we implement a very similar methodology to the generic reservoir operations with the notable exception that the upper and lower operational bounds are not set at 10% and 75% of maximum storage capacity but rather are derived from the observational data using the workflow described in Figure 1. We derive the operational bounds for each type of dam and each location, in other words a fishery dam in the United States can have different operational bounds than a fishery dam in Brazil, yet a hydropower dam in Switzerland could have similar operational bounds to a hydropower dam in Vietnam as long as the input data shows similar trends). For all hydropower dams, we calculate a new initial hydropower release based on equation 7:

$$R_{hi} = \begin{cases} D * RF/B, & \text{if } R < D \\ R & \text{if } R > D \end{cases} \quad (7)$$

where D refers to the maximum demand, R is the currently calculated release, RF is the reduction factor (defined in equation 3, $R_{hi}$ is the initial release, and B is the bankfull discharge.

Once the hydropower release is calculated, there are multiple options for the final release outlined by equation 8. When storage is below conservation, the release is equal to the recalculated environmental flow and the dam meets no downstream demand. When the storage again enters the active zone (the area between the conservation and flood curves), the release is the difference between the current storage ($S_c$) and the hydropower release described by equation 8. If this new release results in a lower storage value than the conservation value, we assume the release is equal to the environmental flow value and does not release any additional water.

$$R_h = \begin{cases} S_c - R_{hi}, & \text{if } S_{min} < S_c < S_{max} \text{ and } S_c - R_{hi} > S_{m}in \\ max(S_c - S_{min}, 0) & \text{if } S_c - R_{hi} < S_{min} \end{cases} \quad (8)$$

where $R_{hi}$ is the initial hydropower release defined by 7, $S_c$ is the current storage, $S_{max}$ is the flood value and $S_{min}$ is the conservation value.





### 2.4.4   Operations for Irrigation-like Dams

For irrigation-like dams, $S_{[min]}$ is set at 10% of the maximum storage capacity and $S_{max}(t)$ is set at the flood value of that given day. When storage is in the active zone and above the dead storage zone (defined as the lower 10%), the release will be
equal to the calculated demand, unless releasing that much water would push the storage into the dead zone (defined as the lower 10% of the storage capacity). In the case where the precalculated release is already greater than demand (1), the release does not change. We use the following piecewise function to capture these dynamics:

$$R_i = \begin{cases} R, & \text{if } S_{min} < S_c < S_{max} \, and \, R > D \\ RF * D) & \text{if } S_{min} < S_c < S_{max} \, and \, R < D \\ max(S_c - 0.1 * S_{cap}, 0) & \text{if } S_c - R < 0.1 * S_{cap} \end{cases} \quad (9)$$

### 2.5   Random Forest Extrapolation

Since the weekly operational bounds from STARFIT are only available for 1752 structures globally, this would severely limit our global modeling capabilities. Therefore, we used a Random Forest approach to extrapolate the 10 parameters for the approximately 22,000 other structures in GeoDAR based on relationships between the parameters for the 1752 dams and their reservoir and catchment characteristics. As input features, we used static reservoir characteristics (i.e. storage capacity and main purpose), socioeconomic variables (i.e. population density), and climatic variables for the upstream areas (i.e. precipitation,
temperature, and aridity) to best reflect potential drivers for changes in reservoir release policies. 75% of the 1752 structures were used to train the random forest model and the remaining 25% were used as independent validation. The obtained RF was then used to extrapolate the 10 parameters to the remaining 24,000 structures.

Based on Steyaert and Condon (2024) and van Beek et al. (2011), we assume that the flood peak cannot be greater than 100% or less than 5% of the total storage capacity. Therefore, any values that sit above 100 or less than 5 for flood are automatically set
at these bounds. We also assume that there is at least a 5% difference between the flood and conservation curves so that there is always an active zone in the reservoir. To ensure this is the case, we calculate the difference between the flood and conservation curves at each weekly step for each dam and if the difference is less than 5, we use equation 10 below to calculate the new conservation level where $C_{[tnew]}$is the conservation value at the current timestep, $F_t$ and $C_t$ are the flood and conservation values at the timestep.

$$[C_{t_new} = min(F_t - 5, C_t)] \quad (10)$$

### 2.6   Model Evaluation and Model Setup

In this study, we use five scenarios to test the impact of different reservoir datasets as well as different reservoir operating schemes on the hydrologic simulations of PCR-GLOBWB 2. Our first two scenarios use the original reservoir operations in



| Model | Reservoir Dataset and number of dams | Operational Policies | Command Area (km) |
|-------|--------------------------------------|----------------------|-------------------|
| Baseline | GRanD (6000) | Sutanudjaja et al. (2018) | None |
| BaseGeoDAR | GeoDAR (24,000) | Sutanudjaja et al. (2018) | None |
| Turn250 | GeoDAR (24,000) | Turner et al. (2021) | 250 |
| Turn600 | GeoDAR (24,000) | Turner et al. (2021) | 600 |
| Turn1100 | GeoDAR (24,000) | Turner et al. (2021) | 1100 |

**Table 2.** Shows the model scenarios that we have built and tested for this paper. The first column denotes the name assigned to the model scenario in the results.

PCR-GLOBWB 2 and only differ by the reservoir input datasets 2: one has the GeoDAR database as the input (BaseGeoDAR)

and one has the GRanD data as input (named Baseline), both of these use the reservoir operating scheme as developed by Sutanudjaja et al. (2018). We then create three scenarios with the updated data-driven algorithm for the three command areas we opted to test based on literature: 250 $km$ (Turn250), 600 $km$ (Turn600), and 1100 $km$ (Turn1100). These three scenarios all use the GeoDAR reservoir dataset as the input dataset and can directly be compared to the Baseline and BaseGeoDAR scenarios that uses Sutanudjaja et al. (2018).

To estimate the impact of different reservoir datasets, we compared the Baseline and the BaseGeoDAR scenarios as any difference in the hydrologic variables (e.g. discharge, reservoir outflow, reservoir storage, reservoir evaporation, surface water abstraction, and total runoff) can be related directly to the different number of reservoirs structures in the model. The comparison between the BaseGeoDAR and Turn250, Turn600 or Turn1100 serves to identify the impact of implementing a different reservoir operating scheme in an existing model like PCR-GLOBWB 2.

To validate our analysis, we compared PCR-GLOBWB 2 discharge at the point location closest to dams where observed discharge is available via GRDC (Federal Institute of Hydrology, 2020). While this validation has its limitations (mainly we are only looking at single point locations and there is a skew towards more gauged basins) it allows us to observe how streamflow regimes are changing between the different simulations, how different that regime change might be between different reservoir release schemes, and to determine which reservoir operation scheme will produce the most accurate streamflow measurements.

To complement this, we also validate our reservoir storage against observed storage values in ResOpsUS as well as remotely sensed reservoir storage from GloLAKES to see if reservoir dynamics are better represented with the new database schemes.

## 3 Results

### 3.1 Impact of reservoir datasets

We first analyze the impact of changing the reservoir dataset on the global representation of reservoirs in PCR-GLOBWB 2.

Figure 2 shows the total storage that is added to the modeling domain when changing from GRanD (global storage of





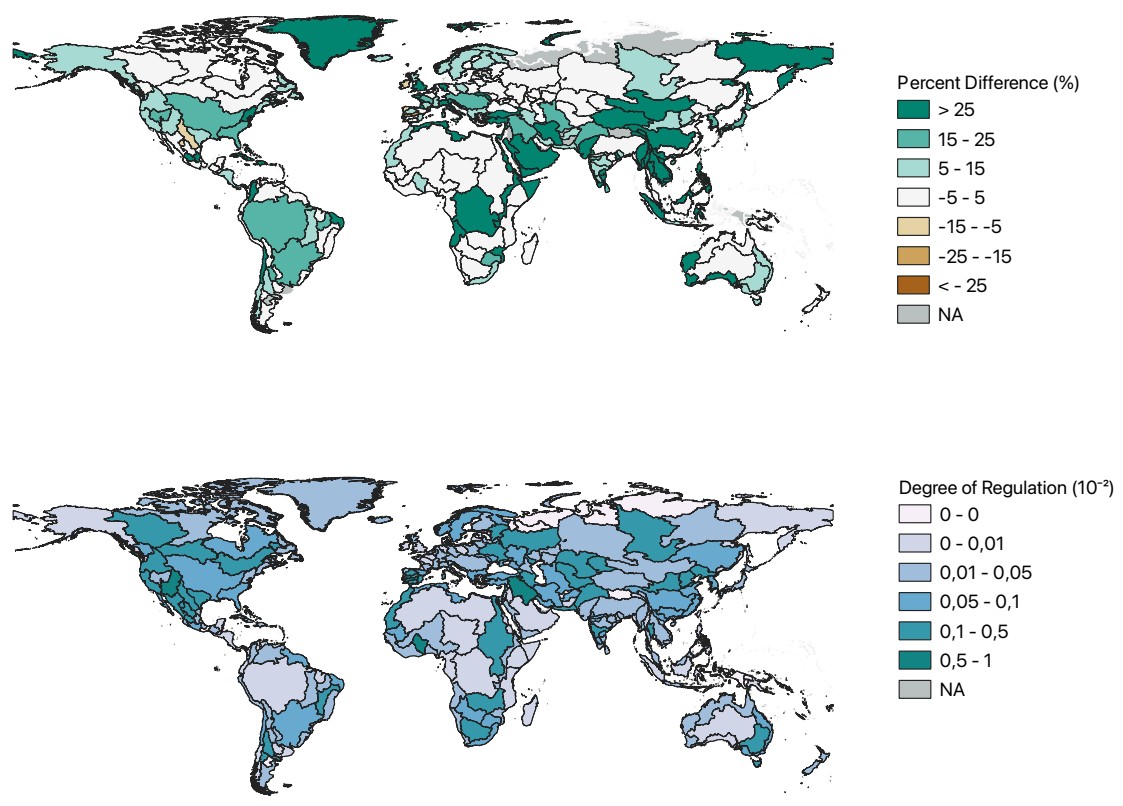

**Figure 2.** Change in total storage globally in GeoDAR when compared with GRanD as a percentage (a) and the fraction of regulation in each basin (b) where grey represents areas that have no storage. Both panels use the PCR-GLOBWBW 2 input files using GeoDAR to determine the total storage in each basin. Panel b uses the average of 40 years of modeled runoff data from PCR-GLOBWB 2.

6,355.72 $km^3$) to GeoDAR (global storage of 7,123.66 $km^3$). 95% of the 230 global basins depicted in HydroSHEDS observe increases in reservoir storage as a result of the inclusion of new dams and only a few basins show a reduction. Of these 218 basins, 40 basins have slight increases in storage (noted as percent differences between 0 and 1%), while the remaining 178 basins observe more significant increases. The largest storage increases are observed in Greenland, Central Asia, the Middle East, the Horn of Africa, and Central Africa. Of the 11 basins that do not observe increases in storage, spread throughout Central Mexico, Brazil, east Africa, China, the Baltic states, Ireland, and Spain, two observe storage percentages very close to 0, and 9 observe much larger negative differences. The lowest value sits at -21% and is located in Ireland, which is the result of corrections in dam locations or storage volumes moving from GRandD to GeoDAR.



These negative values are the result of the two main decisions. The first is the decision to use GeoDAR as the "truth"
value when reporting storage capacity as GeoDAR included GRanD and updated these values where necessary. This decision
could mean that some dams have lost storage capacity due to discrepancies between GRanD and GeoDAR. Secondly, our
workflow for creating the updated input maps for GeoDAR looks at each catchment and determines the total surface area of
each reservoir and lake. Since reservoirs cannot be split across multiple catchments, we calculate the total area of the reservoir
in each catchment and then place the reservoir in the location that has the largest area in large with the HydroSHEDS draining
network in PCR-GLOBWB 2 (Giachetta and Willett, 2018). While this is physically sound, the PCR-GLOBWB 2 drainage
network might not directly align with the physical river networks, or the reservoir dataset does not align with the digital
drainage network, and therefore some reservoir structures might be oriented in a different basin. For small dams, this might not
be an issue, but for larger dams, this could end up causing storage differences.

To evaluate the impact the increased number of dams has on the global streamflow, we aggregated the long-term annual
average runoff from 1979 - 2023 across all 230 global basins and then divided the total storage capacity by the long-term
average annual runoff for each basin. This yields the fraction of regulation per basin depicted in Figure 2b. 20% of the basins
or 42 total basins have zero regulation due to having no reported storage capacity (regions depicted in grey in Figure 2a) while
4 basins have regulation values greater than 0.5, meaning that half of the longterm basin runoff is stored in reservoirs (i.e. in
the southwestern US and the Middle East).

More notable than the exact regulation values are the differences and similarities between the patterns in Figure 2a and
Figure 2b. Basins with a large degree of regulation (shown in Figure 2b) like the Colorado Basin, Yenasei, and the Tigris-
Euphrates have a large amount of storage which suggests that these regions have a multitude of medium to large dams compared
to the available runoff. Conversely, basins with a large amount of storage (Figure 2a) such as much of Central and South Eastern
Asia, Central Africa, and Western Australia do not have a high degree of regulation (Figure 2b).

**3.2  Evaluation of the Random Forest Model**

First, we added the 24,000 structures to the PCR-GLOBWB 2 domain and developed the workflow described in Figure 1 to
incorporate two main types of reservoir operations into PCR-GLOBWB 2. We then evaluated the impact of this methodology
by using different input datasets: ResOpsUS (n=668, Steyaert et al., 2022) and GloLakes (n=24,000, Hou et al., 2024) in
four different combinations. The first combination is a comparison between the different STARFIT curves obtained from each
reservoir dataset (Table 3: column 1). This comparison (using 668 dams across the contiguous United States) allows us to
evaluate the implicit impact of using local data (ResOpsUS) compared to a global remotely sensed dataset (GloLakes), that is
required for global applications like global hydrologic model simulation. The difference we find in this comparison shows the
potential reduced quality of global reservoir water level estimates compared to local information. The second combination is
a comparison between the STARFIT-derived curves for ResOpsUS (Table 3: column 2) and the extrapolation of those curves
(n=668) using our random forest methodology in Section 2.5. This allows us to evaluate how well our methodology matches
the original derivation of reservoir operating curves compared to the benchmark dataset of STARFIT. The last two comparisons
are between two different data products from GloLakes for satellite altimetry: IceSat and LandSat (Table 3: columns 3 and 4),





| Performance Metric | STARFIT with Glo-Lakes vs STARFIT with ResOpsUS (n = 266) | STARFIT vs RF extrapolation with ResOpsUS (n = 668) | STARFIT vs RF extrapolation with GloLakes/Sentinel (n = 1753) | STARFIT vs RF extrapolation with GloLakes/ICESAT (n = 2445) |
|---|---|---|---|---|
| Bias Flood | 11.58 | 0.66 | 0.16 | 1.977 |
| Bias Conservation | -8.05 | -3.06 | 0.08 | 1.38 |
| Correlation Flood | 0.83 | 0.78 | 0.96 | 0.98 |
| Correlation Conservation | 0.59 | 0.73 | 0.94 | 0.97 |
| RMSE Flood | 18.03 | 20.16 | 11.31 | 8.91 |
| RMSE Conservation | 15.46 | 18.32 | 10.13 | 7.47 |

**Table 3.** Depicts the performance metrics (column 1) between the extrapolated curves using the methodology in Figure 1 and the STARFIT algorithm developed by Turner et al. to evaluate the impact of changing data and the random forest on the "original curves." All RMSE values are in the units ($percent/week$). Column two shows the metrics between the original curves derived from STARFIT and the constrained curves using 668 actual storage values from ResOpsUS (Steyaert et al., 2022) in order to evaluate the impact of using Glolakes. Column three depicts the RF workflow vs the extrapolation using ResOpUS to further evaluate the RF workflow without the error in satellite altimetry. Column four depicts the comparison between the STARFIT curves and the random forest-constrained curves for the 1752 GloLakes dams that could be input into STARFIT and were used to train and validate the RF workflow. Lastly, column five shows the same metrics for another data product in GloLAKES using IceSAT. This is used to validate the RF workflow.

which allow us to see the difference in quality moving from local to global and using a random forest compared to the original methodology.

The bias metrics in the first column show that STARFIT with Glolakes overestimates the flood level while the conservation level is slightly underestimated, yet the higher correlation demonstrates the timing is consistent between the two datasets. Conversely, the conservation correlation is not as strong (0.59), however, the conservation curves are slightly closer to the original STARFIT data than the flood curves.

After evaluating the uncertainty of the input STARFIT curves from direct observations and the satellite altimetry data (Ta-

ble 3: column 1), we evaluate the extrapolation methodology in Figure 1 with ResOpsUS (Table 3: column 2). The third column shows the bias between the random forest algorithm and the original algorithm. Overall, we observe a large bias for the flood curves and quite high correlations for flood and the conservation curves suggesting the RF workflow is overestimating the flood curves yet still aligns with the values in STARFIT. RMSE for both the flood and conservation curves are high which demonstrates that while the timing of the flood curves match, the general errors are much larger in part because the RF algorithm

underestimates the conservation levels. Compared to column two, the conservation correlations rise and the bias decreases which suggests the random forest algorithm trained on observational data does a relatively good job of capturing the levels for the conservation curves compared to the STARFIT algorithm. The RMSE values do increase and the flood correlation de-



creases suggesting that the RF does not accurately capture the flood values potentially due to the short notice period for flood conditions.

Columns 3 and 4 show the extrapolation metrics for both GloLakes datasets: IceSat (column 4) and Sentinel (column 3). These results show that when applied to GloLakes, the RF extrapolation effectively reproduces the flood and conservation curves, even better than when the RF is applied to the ResOops STARFIT curves (Column 2). However, column 1 shows that the largest error occurs when moving from ResOpsUS to GloLakes. Thus the total errors are mostly made up of errors in the storage time series that are obtained from remote sensing. In general, the operating bounds are more narrow compared to the

original algorithm (Figure 3).

While the metrics depicted in Table 3 provide a summary and validation of our workflow, the actual curves related to the different data and method combinations are much more insightful. For illustration, Figure 3 shows the results for dams with two main purposes: the Clinton Lake Dam (Illinois, United States) with a hydropower main purpose (Figure 3 top row) and the Medina Dam (Texas, United States) with an irrigation main purpose (Figure 3 bottom row). We only include *hydropower-like* and *irrigation-like* as our allotted reservoir operations in Section 2.4 use these two categories to determine operations.

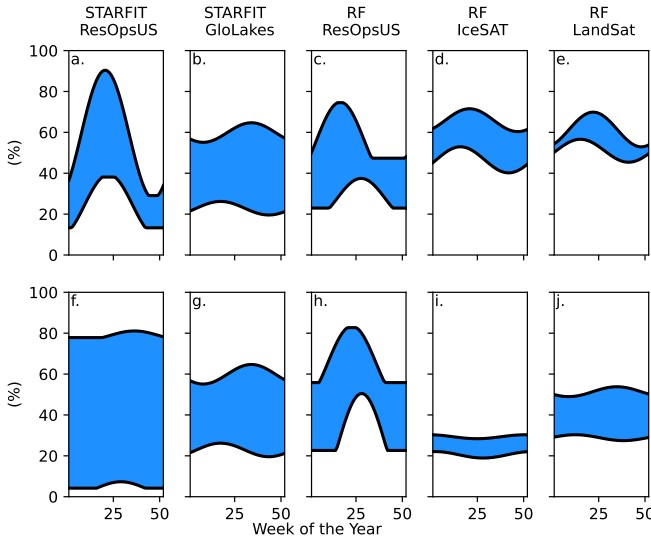

**Figure 3.** Operational curves as storage percentage vs epiweek for two main types of dams: hydropower (a, b, c, d, e) and irrigation (f, g, h, i, j). Panels a and f depict the original curves that come out of the STARFIT model. Panels b and g depict the STARFIT model using the data from GLoLAKES LandSat. Panels c and h depic the Random Forest algorithm trained on ResOpsUS. Panels d and i depict the Random forest methodology shown in Figure 1 using the GLoLAKES IceSAT data, while panels e and j show the final constrained curves from the random forest methodology using GloLakes LandSat.


When looking at the hydropower example, Clinton Lake Dam, (Figure 3: top), we observe that all the curves have a peak during the first half of the year which aligns with spring precipitation in this region. The ResOpsUS (Figure 3a) curve has a larger peak (a range of 20% and 95 %) when compared to the other algorithms. GloLakes-STARFIT (Figure 3b), on the other




hand, has a much more limited operational range (between 20% and 75%). This range is further limited when looking at the
RF model using GloLakes-IceSAT or GloLakes-LandSat (Figure 3d and 3e respectively), although these final storage ranges
encompassed the average range of both ResOpsUS and GloLakes. The RF model with the ResOpsUS data (Figure 3c) is more
similar in range to the STARFIT curves made with GloLakes while maintaining the same peak in the early part of the year.
This suggests that in general, our RF workflow constrains the active zone of the reservoir while keeping the same seasonal
patterns when compared to the STARFIT curves for the hydropower-like dams.

For *irrigation like* dams, we plotted the same five curves for the Medina Dam (irrigation main purpose). Overall, the curves
depict the same general trend (Figure 3 bottom) with the ResOpsUS (Figure 3f) curves having the largest active zone and
the RF workflow with GLoLakes-IceSAT (Figure 3i) having the most limited operational range. Opposite to the hydropower
example, the final GloLakes-LandSat extrapolation (Figure 3j) shows a much lower overall active zone with slight increases in
the active zone towards the latter part of the year, which is consistent with irrigation demand. In general, we see that moving
from local data ResOpsUS to more global data like GloLakes results in the strongest reduction in the operation bounds, while
moving from STARFIT to RF only results in a small reduction of the operational bounds.

To evaluate the implication of these operational bounds, we first look at the impact on two dams: the Clinton Dam in Illinois
(with a water supply main purpose) and the Koelnbrein Dam in Austria (with a hydropower main purpose) shown in Figure 4.
To do this, we plot the monthly average storage fraction (Figure 4a and 4d), the difference between the reservoir inflow and
outflow at the point location of the dam (Figure 4b and 4e) and the discharge in $m^3/s$ at the respective basin outlets. For each
panel, we plot three of the five models: Baseline (black, the reservoir rules as currently implemented in PCR-GLOBWB 2),
BaseGeoDAR (grey, original reservoir operating rules with additional reservoirs), and Turn250 (pink, new reservoir rules
and additional dams). We opted to not plot all Turn600 and Turn1100 as difference between the models are relatively small
(Appendix Figure A1).

In both cases, the reservoir storage is lower in the data-driven operations (Turn250) when compared to the generic oper-
ations (Baseline and BaseGeoDAR). This is mostly due to the change in operational schemes but also affected by upstream
regulation and changes therein (the difference between Baseline and BaseGeoDAR). For the hydropower dam, the Baseline
and BaseGeoDAR storage fractions are not too different and have the same seasonal cycle. Conversely, the Turn250 storage
fractions sit between 0.01 and 0.02 with a shifted storage fraction peak towards the end of the year compared with the Baseline
and BaseGeoDAR which have a storage peak towards the spring. The water supply dam, which contains irrigation-like oper-
ations, shows similar seasonal trends as the Baseline and BaseGeoDAR, yet the average storage drops much lower, especially
in the autumn and winter months.

To fully determine the shifts in storage observed in Figure 4a and 4d, we can use Figure 4b and 4e to determine when the
reservoir is filling (positive values) and when it is emptying (negative values). In all models, the irrigation dam fills in the winter
and spring and empties during the summer. The Turn250 model has slightly more filling, which is offset by more depletion
in the spring months, indicating larger storage dynamics, but a lower average storage. It also does not return to full quite as
quickly as the Baseline and BaseGeoDAR models, potentially due to meeting (downstream) demand. For the hydropower dam,
all models have a peak in the springtime with a decrease in all other months. Conversely to the irrigation dam, the Baseline





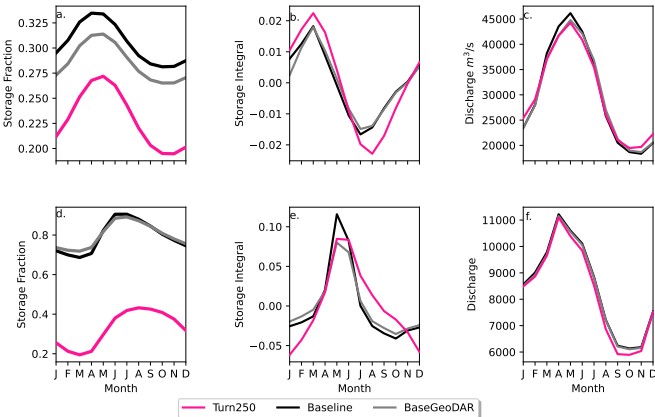

**Figure 4.** Implementation of the final operational curves in PCR-GLOBWB 2 for the three main reservoir models: Baseline (black), BaseGeo-DAR (grey), Turn250(pink). The top row shows Clinton Lake, a water supply dam in the Mississippi basin, while the lower row shows the Koelnbrein dam, a hydropower dam in the Danube basin. Column one (panels a and d) shows the storage fraction for all the models. The second column (panels b and e) shows the change in storage between each month to observe where the dams are filling (positive values and releasing (negative values). The final column (panels c and f) shows the discharge at the respective basin outlets.

model has more filling compared to the BaseGeoDAR and the Turn250 models. That said, the springtime depletion in the
Turn250 model is much more linear compared to the other models.

The different reservoir schemes have small impacts at the basin scale, even in heavily managed basins (Figure 2). In the Mississippi (Figure 4c), there is a small reduction in discharge for the Turn250 model during the peak flows in the spring and slightly more discharge during the low flows in the winter months. In the Danube (Figure 4f), there is slightly less discharge at the outlet during the winter for the Turn250 model, but otherwise, the modeled discharge is the same. The difference between
these two basins is in part due to the difference in regulation observed in Figure 2b, where the Danube basin has a lower degree of regulation compared to the Mississippi basin.

### 3.3 Reservoir model comparison and implications on storage and discharge

To validate the implementation of the new reservoir scheme, we first calculate the different water balance components (Table **??**). The data-derived operations do not affect the climatic forcings; therefore, we do not observe differences in precipitation
and only small differences in evaporation across all models. There is an increase in evaporation in the BaseGeoDAR model and an increase in water availability (denoted by an increase in water body storage and a positive change in total water storage across the model time frame). These increases as well as the increase in surface water abstractions are a potential result of increasing the total number of dams and the total storage capacity. Comparatively, we observe a decrease in water availability (denoted by water body storage, change in total water thickness and runoff) as well as decreases in water body evaporation and
runoff in the data-derived operations (Turn250, Turn600, Turn1100) a potential result of the lower storage levels. Overall, the





| Model | Precipitation ($km^3/yr$) | Evaporation ($km^3/yr$) | Runoff ($km^3/yr$) | Surface Water Abstraction ($km^3/yr$) | Groundwater Abstraction ($km^3/yr$) | Water Body Storage ($km^3/yr$) | Water Body Evaporation ($km^3/yr$) | Change in Total Water Storage ($km^3/yr$) |
|---|---|---|---|---|---|---|---|---|
| Baseline | 62,803.44 | 111,556.56 | 49,029.85 | 2,467.25 | 566.18 | 10,570.41 | 1,651.557 | -6.94 |
| BaseGeoDAR | 62,822.95 | 111,556.56 | 49,049.07 | 2,499.56 | 557.91 | 10,561.87 | 1,657.99 | 6.934 |
| Turn250 | 62,794.48 | 111,556.56 | 49,021.39 | 2,519.99 | 561.670 | 8,614.62 | 1,627.17 | -4.84 |
| Turn600 | 62,794.53 | 111,556.56 | 49,025.71 | 2,518.20 | 561.44 | 8,614.97 | 1,627.23 | -4.78 |
| Turn1100 | 62,794.78 | 111,556.56 | 49,021.83 | 2,520.12 | 561.67 | 8,614.55 | 1,627.23 | -4.70 |

**Table 4.** Different hydrologic components for evaluating the water balance across all the models. In addition to hydroclimatic variables (precipitation, evaporation, and total water storage), we also include storage in lakes and reservoirs, evaporation from waterbodies, and abstractions from ground and surface water.

water balance results demonstrate that the data-derived operations do not lead to large differences across the PCR-GLOBWB 2 domain, suggesting that the relative impacts are not large and are mostly regional to local.

To further validate the data-derived operations, we analyze discharge and reservoir storage, as the water balance table shows that both are affected by changes in reservoir operations. First, we validate the discharge in our models against the observed values in GRDC. Globally, 6,044 stations fit our criteria: a period of record starting in at least 1979, a minimum overlap of two or more years, catchment sizes that are at least 25% of each other, and no more than a difference of three magnitudes between the observations and the simulated discharges. As other reservoir studies demonstrate that reservoir impacts on streamflow are greatest near the dam (Hanasaki et al., 2006; Haddeland et al., 2006; Biemans et al., 2011; Zajac et al., 2017), we filtered out any locations that were not directly impacted by upstream reservoirs. This left 2,666 gauges that fit the above criteria. For these gauges, we calculated the Kling-Gupta Efficiency (KGE) and its components (correlation, bias ratio, and variance ratio) for each location and each model and plotted a cumulative distribution function (CDF, Figure 5). The CDF shows slight improvements in the middle of the KGE range (-0.25 to 0.25), however, the improvements are quite minor. This means that our reservoir operations do not lead to significant improvements in the discharge simulations in PCR-GLOBWB 2 at the measurement locations and as a result do not allow draw conclusion into the impact of data-derived operations on streamflow regimes.

To identify the impact of the two reservoir operation schemes, we calculated the differences between the individual components of KGE (R, beta, alpha). We included the current operational scheme in PCR-GLOBWB 2 with the inclusion of GeoDAR (BaseGeoDAR) and the data-driven operations for the 250 command area (Turn250) compared against the Baseline model (Figure 5b, c, and d respectively). We decided not to include the other command areas as a boxplot of the KGEs per command area (i.e. Turn250, Turn600, and Turn1100) depicted no variations (Appendix Figure A1). When looking at the differences in the correlation (Figure 5b), there are more positive correlations for BaseGeoDAR and more negative correlations for Turn250 when comparing these two models with the Baseline (42.78% above 0 for the BaseGeoDAR operations vs 41.46%




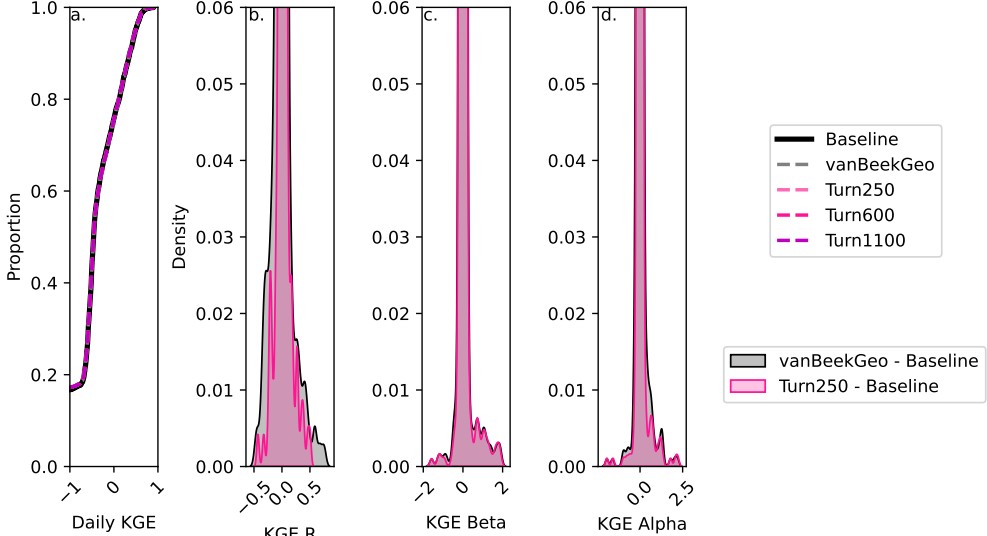

**Figure 5.** Depicts the validation between the five models and the GRDC streamflow gauges globally. Panel a shows the daily KGE values for the 2,666 stream gauges that have at least one dams upstream and are therefore directly impacted by our reservoir model. Each line is colored by the model. Panel b, c, and d show the difference between the baseline for the BaseGeoDAR and Turn250 models for the R (b), alpha (c), and beta (d) components of the KGE.

for Turn250 respectively). This suggests that in some cases, the inclusion of more dams is enough to improve model performance, but in most, it is not. However, the magnitude of these differences is quite small as there are only four points above or
below a difference of +/-0.5 for the comparison between Turn250 and the Baseline and one point for the comparison between BaseGeoDAR and Baseline. For the bias difference (Figure 5c), the Turn250 model contains more bias values above and below 0 (50.86% vs 35.29% above and below for Turn250 and 52.22% vs 35.15% for the BaseGeoDAR). This suggests that the data-derived scheme is more likely to overestimate discharge than the generic operations with the GeoDAR maps. Lastly, the alpha (Figure 5d), which depicts variance ratio of the observations and modeled values, depicts more variance in the Turn250
model when compared to the Baseline (33.56% vs 40.78% above for Turn250 and 44.78% vs 37.56% for the BaseGeoDAR). This suggests that data-driven operations are better at representing the variability of discharges.

In addition, we validated the reservoir storage derived from each operational scheme against indirect observations (GloLakes) and direct observations (ResOpsUS, Figure 6a and 6b respectively). To do this, we compared the long-term storage fraction for the Turn250 (depicted in pink) and Baseline (depicted in black) models against the two observations of reservoir levels. In
both panels, the data-driven operations (Turn250) are more closely aligned with the observations (i.e. sit close to and above the 1:1 line) than the Baseline operations (Figure 6). Figure 6a shows the scatter between indirect observations (reservoir levels are translated to reservoir storage) taken from satellite altimetry using GloLakes. Here, we observe that the Turn250 aligns more with GloLakes (an RMSE of 0.28 and more points at or above the 1:1 line) than the Baseline operations (an RMSE of 0.32).





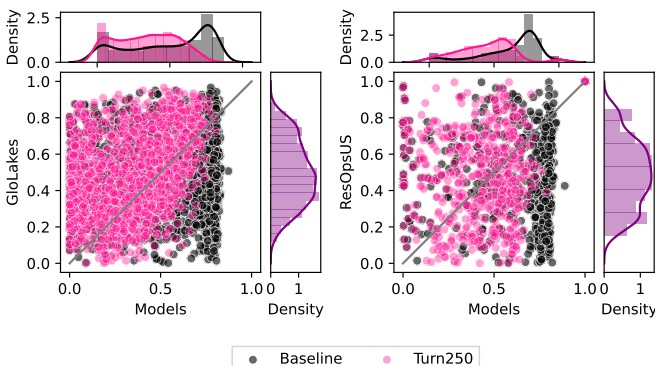

**Figure 6.** Shows scatter plots of reservoir storage observations vs modeled results for Turn250 (pink) and Baseline (black). Panel a uses GloLakes as the observations, which is the dataset the operational policies are derived from) while panel b uses ResOpsUS as a more "neutral" validation. For both panels, we matched the same period of record as GloLakes (01-01-1984 to 05-31-2023) and ResOpsUS (variable depending on the reservoir, but typically 1980 - 2020).

| Model | RMSE with GloLakes | Correlation with GloLakes | P-value with GloLakes | RMSE with ResOpsUS | Correlation with ResOpsUS | P-value with ResOpsUS |
|---|---|---|---|---|---|---|
| Baseline | 0.32 | 0.18 | $8.53 * 10^{-14}$ | 0.37 | 0.08 | 0.06 |
| Turn250 | 0.28 | 0.37 | $4.78 * 10^{-53}$ | 0.30 | 0.14 | 0.0008 |

**Table 5.** Shows the RMSE, the spearman rank correlation and corresponding p values, between the scatter plots in Figure 6 for each of the models (rows) and between the two datasets(columns). In both comparisons, the correlations are statistically significant.

Next, to remove the underlying bias that we trained our RF algorithm on data in GloLakes, we also compared our data-driven operations to ResOpsUS. Figure 6b shows a scatter of the modeled storage fractions compared to observations for large dams in the United States (ResOpsUS). Once again, the Turn250 model aligns better with observations (RMSE of 0.30 for Turn250 vs 0.37 for the Baseline). Additionally, the Spearman rank correlation between both observations and models is moderately strong for Turn250 when compared to GloLakes (0.37 for Turn250 and 0.18 for Baseline (Table 5 and slightly positive for ResOpsUS (0.14 for Turn250 vs 0.08 for Baseline). Further analysis of the differences between the two main types of reservoirs in our analysis (*irrigation like* and *hydropower like*, demonstrate that our hydropower operations are more similar to the observations (RMSE of 0.28 for Turn250) compared to both the generic operations (RMSE of 0.32) and the *irrigation like* dams (RMSE of 0.29 for Turn250) when comparing against the GloLakes dataset (Figure A2). In conclusion, this analysis demonstrates that the data-driven operations have more realistic long-term average storage compared with the Baseline scenario.

To round off our analysis, we look at the density of all the storage fractions (Figure 6) for GloLakes and ResOpsUS. For both of the observations, the majority of the observed storage fractions sit between 20% and 80% full slight variations in the exact peaks (30% full for GloLakes and 50% full for ResOpsUS). The storage fractions from the Turn250 operations contain





more storage fraction values between 20% and 60% and in doing so underestimate the total storage when compared to both observations. The Baseline storage fractions, on the other hand, contain a large density of values between 60% and 80% full, which suggests that the Baseline operational scheme holds reservoirs at a higher storage value on average and is overestimating the amount of storage. Ultimately, we find that the storage estimates in the Turn250 model are more accurate when looking at the correlations and RMSEs.

### 3.4 Reservoir Regulation Changes Over Time

To determine regions where the Turn250 model is able to capture storage dynamics, we opted to look at the median long-term monthly reservoir fraction for each basin in HydroSHEDS and the aggregation per continent (Figure 7). In addition to plotting the monthly storage fraction of all the models, we plot the monthly storage fraction for the GloLAKES-LANDSAT+Sentinel2 reservoirs (the training dataset for our RF workflow) to allow for comparison between observations and model results. 468 dams had a storage fraction greater than one when using the static values from our GeoDAR input maps, therefore, for these dams, we used the maximum storage value in the GloLakes observations as the maximum storage for this analysis. This misalignment could be due to flood conditions in the GloLAKES reservoirs or overestimations due to the workflow in Hou et al..

In looking at the median storage fraction across the different basins (Figure 7), we first observe higher storage fractions in the more northern basins. Basins with a large amount of regulation such as the Mississippi, Nile, and Orinoco (based on Figure 2) have median storage fractions slightly above 0.5. The global differences in median storage fraction generally align with increased aridity, where regions that are more arid such as Australia, the Sahara, Mexico, the Middle East, and India have lower median storage fractions compared with the potential demand for water supply in these regions. Conversely, more humid regions such as Northern Europe, the Amazon, the Mekong, and the Eastern United States have median storage fractions around 0.4.

Globally, the generic operations hold more water as the Baseline and BaseGeoDAR have the higher median storage fraction of 0.63 and 0.59 respectively. Turn250 (0.50) sits closer to the median storage fraction observed in GloLAKES (0.48). Regionally though, the differences are more pronounced as the data derived operations do not capture the storage fractions observed in North America, Australia and South America, while in Europe, Asia and Africa the data derived operations align better with observations. In general, though, all the models (Turn250, BaseGeoDAR and Baseline) have similar monthly trends that do not necessarily align with the monthly trends in GloLAKES. For example, in North America, the observed storage values are highest in March through July, while Turn250 has the highest storage in August to October and the Baseline and BaseGeoDAR models both have fairly flat monthly storage values. Conversely, in Asia and South America, the modeled monthly storage fraction (specifically when looking at Turn250) trend appear to align relatively accurately with the observed storage fractions.



**Figure 7.** Median storage fraction per basin in HydroSHEDS for the Turn250 as a spatial map surrounded by line plots. Each line plot corresponds to a different continent and shows the long-term monthly storage for each model: Baseline(black), BaseGeoDAR (grey), Turn250 (pink), compared with the long-term monthly storage fraction in GloLAKES (navy)

## 4 Discussion

### 4.1 Global Impacts on the Hydrologic Cycle of the Data-Driven Reservoir Operations

To analyze the impact of the reservoir operations on the global hydrologic cycle, we first analyzed the ability of the model setup to reproduce streamflow dynamics. Primarily, we evaluate the cumulative distribution of the daily KGE values, which shows little to no difference between the different model configurations (Figure 5a). This is partly due to the limited number of GRDC gages (only one-third) that are directly downstream of reservoirs and thus the ability of this validation to directly quantify the impact of changes in reservoir operations. While the correlations are not affected by the changes in reservoir operations, we do observe that the variance ratio shows (Figure 5d) that the data-derived operations are slightly better at capturing the variability





| Region | Baseline RMSE | BaseGeoDAR RMSE | Turn250 RMSE |
| --- | --- | --- | --- |
| North America | 0.32 | 0.24 | **0.21** |
| South America | **0.05** | 0.05 | 0.15 |
| Europe | 0.09 | 0.12 | **0.05** |
| Australia | 0.31 | **0.08** | 0.22 |
| Asia | 0.18 | 0.10 | **0.06** |
| Africa | 0.18 | 0.07 | **0.04** |
| Global | 0.11 | 0.15 | **0.04** |

**Table 6.** RMSE between the curves in Figure 7 for each of the given models (columns) and regions (rows). For this table, we took the GloLAKES storage fraction as the observations. For clarity, we depict the best results in bold.

of the streamflow dynamics and the bias ratio shows the data-derived operations are more likely to overestimate streamflow
(Figure 5c). The sensitivity of the two operational schemes to the different components of KGE is most likely a result of the localized impacts which are more pronounced when we look at the difference between the long-term reservoir storage integral (Figure 4b and e). While the aggregated results tend to dampen the impact of the reservoir operations on the streamflow dynamics (Figure 4c and f, and Figure 5a), we see that locally there are still distinct differences in the simulated reservoir outflows. In irrigation-like dams for the data driven operations, the magnitude of release is larger than that of inflow during
the autumn months when compared to the generic schemes(Figure 4c). This suggests that the addition of downstream demand into our reservoir scheme increases the overall drawdown of the reservoir and allows more water to move through the system, a dynamic Voisin et al. (2013) observed in their scheme.

Due to the limitations in monitoring reservoir impacts on global streamflow dynamics, we recommend using alternative methods such as regional or, where they exist, global reservoir storage observations. Compared to the streamflow results which
do not show strong impacts, we observe that the data-derived storage is more aligned with observations and therefore provides a more accurate reservoir storage representation globally (Figure 7 and Figure 6). In most regions except Australia, the reservoir storage in the data-driven operations is decreased when compared to the generic operations (Figure 7). These lower storage values are most likely due to the transference from GloLAKES to the final curves using the methodology in Section 2.5 as this transference led to more constrained operational bounds compared to ResOpsUS (Figure 3). We also observe the data
derived operations do not align as well as the Baseline or BaseGeoDAR for Australia and North America (Figure 7). This could be due to data gaps in the GloLAKES dataset for these regions, more hydropower reservoirs, or operational patterns that have shifted due to recent drought events, which our random forest workflow may not be able to capture as well as it does not include temporal evaluations of the operating boundaries. Our model water balance shows less reservoir storage, evaporation, and more surface water abstraction in the data-driven operations (Table 4). This suggests two things: first the
water is moving more quickly through the river system as there is not as much storage compared to the generic operations (Baseline and BaseGeoDAR) which results in more available water for abstraction at downstream locations (shown by the difference in surface water abstractions between Turn250 and Turn1100). While this could create a water deficit in locations





directly near the reservoir, the comparison with observed data in Figure 6 demonstrates that the lower storage values are more comparable to independent observations than the larger storage values in the generic operations (Figure 6). Therefore, this

suggests that the generic operations overestimate the amount of water in storage and are not accurately pinpointing regional or localized water deficits (Salwey et al., 2023; Steyaert and Condon, 2024).

   While using observed storage values is ideal for training reservoir models, the global datasets are limited by limitations in data coverage and privacy concerns (Steyaert et al., 2022; Salwey et al., 2023). Therefore to capitalize on our global methodology, we utilized satellite altimetry data that correlates reservoir surface area with reservoir storage, instead of using observed

storage values directly. Ultimately, we observed that satellite altimetry data did not reduce the overall quality of our storage results in many regions, however, the GloLakes dataset does overestimate storage in 468 of the 1752 dams in our analysis when using storage capacities from GeoDAR. While this skews the seasonal trends observed in Figure 7, it is a result of a larger issue in data availability for reservoir management schemes. Since our analysis utilizes the general trends and dynamics in the observations to derive trends of storage fraction that are combined with reported storage capacities in GeoDAR to derive

releases in PCR-GLOBWB 2, it is hypothesized that satellite altimetry is less able to capture the conservation dynamics (Hou et al., 2024; Zhang et al., 2020). However, we observe that the lack of seasonality in the conservation curves allows the random forest algorithm to better represent these curves when compared to the flood curves (RMSE of 7.47 vs an RMSE of 8.91 for the analysis with GLoLAKES-IceSat in Table 3). The random forest algorithm trained on GloLakes has high correlations between the values from STARFIT further suggesting that the use of satellite-derived storage dynamics is the largest area of uncertainty

in our modeling framework. The biggest change is the decreased active zone both in range and in maximum and minimum values (Figure 3 and Figure 6). This is because satellite altimetry data uses storage-area relationships that tend to regress to the mean and that may not necessarily account for natural processes that decrease the overall amount of available storage, such as sedimentation, and are limited in their temporal and spatial resolution. Conversely, observed storage time series such as ResOpsUS use the actual storage values derived from simplified water balance equations or storage elevation relationships and

contain a much larger active zone (Figure 3). This decrease in the operational range will limit the amount of water that can be stored in reservoirs and as a result, will increase the reservoir release (Figure 4b and 4e). However, we still observe through Figure 6 that the data derived storage values are more accurate than the storage values obtained from the generic operations even with the underestimations in part because the generic operations are not as sensitive to observed changes.

### 4.2    Evaluation with Other Reservoir Models

Similar to other modeled reservoir operations, we observe limited improvement in discharge at basin outlets (Hanasaki et al., 2006; Haddeland et al., 2006; Biemans et al., 2011; Zajac et al., 2017), yet the data-derived storage improvements are much more similar to those of Turner et al. (2021) when using the extrapolated curves. We also observe more storage values aligning with observations for different reservoir functions. Salwey et al. (2023) noted that grouping reservoirs based on main purpose may miss key differences in the operational patterns, however, our results demonstrate that the hybrid incorporation of data-

driven operational bounds with a more generic reservoir operation scheme (grouped by two main categories) performs better than the generic schemes previously used. Ideally, we would be able to differentiate into more specific reservoir categories;





however, at the global scale operational information for all main purposes (primarily for fisheries and recreation) is missing. We also find modest improvements in all major reservoir use categories (irrigation, hydropower, water supply, and navigation) by separating the values in Figure 6 based on the main purpose (Figure A2). Additionally, we also observe that our modeled
reservoir storage aligns with the monthly trends Steyaert and Condon (2024) found in the United States (i.e. flood and navigation reservoirs sit much closer to a storage fraction of 0.3 while irrigation-dominated basins have a higher storage fraction with large seasonality). While the trends in the United States are not necessarily indicative of operational trends globally, this is one of the few studies that look directly at the trends seen in storage observations instead of solely looking at the operational impacts.

This said, our model provides a few notable improvements over other reservoir operation schemes. First and foremost, it utilizes observational data to derive key boundaries for reservoir operational bounds, something, that to our knowledge, has not yet been done on the global scale (Turner et al., 2021; Yassin et al., 2019). The other reservoir schemes to utilize observational data directly typically require multiple parameters (such as in Yassin et al.; Turner et al.; Burek et al.) or thus are parameterized regionally (Turner et al., 2021; Salwey et al., 2023; Brunner and Naveau, 2023; Macian-Sorribes and Pulido-Velazquez, 2020).
Generic operations are typically chosen for global hydrologic models due to the ease of incorporation that relies on limited data (Haddeland et al., 2006; Hanasaki et al., 2006; van Beek et al., 2011; Sutanudjaja et al., 2018). Our scheme is scalable both to specific regions and data availability and also to model complexity. This is because the main variable of importance is the derived operational targets which can be easily obtained from limited remotely-sensed observational data.

Recently, there has been a strong push to back-calculate reservoir impacts from readily available hydrologic variables such
as streamflow (Salwey et al., 2023; Brunner and Naveau, 2023). These schemes show promise in areas where reservoir data is not readily available and in models that are currently relying on generic operations based on two main purposes: irrigation or hydropower (Salwey et al., 2023). However they still rely on the assumption that reservoir releases are the difference between naturalized streamflow and observed values (Brunner and Naveau, 2023). This can be accurate in many cases, but in human-dominated basins with a large amount of surface water abstraction and complex interactions, such as the Colorado River basin
among others, this assumption may not be valid. Salwey et al. (2023) remedied this with their transfer function approach, however, the results were only tested in water supply-dominated basins in the United Kingdom, and different balances of uses and operational patterns could change the results. Finally, in all instances, these methods assume that reservoir operations are static over time which may not necessarily be the case (Patterson and Doyle, 2019; Patterson et al., 2021) and would require nearby downstream observations of river discharge to infer the reservoir operating policies over time. While this is not
infeasible there are currently a limited number of reservoirs in the world where this would be a viable approach to accurately derive reservoir operations.

Utilizing data-driven operational bounds combined with a dual-purpose operational scheme has a plethora of benefits (more realistic operational bounds, the ability to change operations based on location, use, and hydroclimatic variables, and increased understanding of water availability). That said, there are still improvements to be made. First, the lack of global reservoir
storage observations or even regional data hindered our ability to create target curves based on observations and that do not rely on satellite altimetry data and its uncertainties (temporal gaps, back-calculated storage, etc). Additionally, we assume



that our operational bounds are constant during the simulation period, however in a changing climate this assumption may no longer hold, especially during extreme events (such as droughts) or under management changes (Salwey et al., 2023; Patterson and Doyle, 2019; Steyaert and Condon, 2024). We also assume reservoirs are operated as a single entity and do not explicitly and dynamically account for releases in series as we assume that the underlying storage data accounts for this dynamic and therefore these relationships exist in our operational bounds.

Based on these limitations, we find that more effort needs to be put into cultivating global datasets of historical reservoir operations and also include more data-derived methods in operational reservoir schemes. While the first recommendation is liable to privacy concerns (Steyaert et al., 2022), we believe that more accurate information on reservoir operations will increase the reliability of large-scale hydrological models and thus serve a wider community. The addition of more data-derived methods in our reservoir operations scheme demonstrated much lower storage than PCR-GLOBWB 2 had previously modeled using other methods. This could highlight unknown water deficits as the total reservoir storage in select regions may not be able to account for surface water abstractions and the currently implemented reservoir operating rules could overestimate the amount of available water in vulnerable regions.

## 5 Conclusions

This study combines previous work by Turner et al. (2021) and data acquisition by Hou et al. (2024) to develop a workflow for implementing data-driven reservoir operations in global hydrologic models. Using an updated dataset of reservoir locations (Wang et al., 2022), satellite altimetry data from Hou et al. (2024) and the STARFIT model (Turner et al., 2021), we first developed operational bounds for 1752 reservoirs whose storage time series were estimated using remote sensing that are then used to train a random forest algorithm. We observe that the RF extrapolation is accurately able to depict the flood and conservation curves and that the main source of uncertainty is the errors associated with the storage estimations from satellite altimetry. After this evaluation, we then estimated operational bounds for over 20,000 structures that are implemented into PCR-GLOBWB 2 using the trained random forest algorithm. While the impact on discharge is modest and does vary significantly based on location, we see large improvements in our reservoir storage and seasonal dynamics when using data-derived operations over the generic ones. Therefore, we suggest reservoir operation models rely primarily on validation of storage in place of validation solely on streamflow as the available streamflow observations are rarely close to the release point of the reservoir and therefore not as sensitive to reservoir operations compared to storage. We also do not see strong differences when using different command areas even though it would be expected that changes to these values would increase or decrease storage. This suggests that reservoir modelers can opt to use the command area that makes the most sense to their domain (we opt to use the 250). To remedy issues with the static operations (i.e. no change in the future), in the future we hope to evaluate changes in operational patterns using smaller 10-year moving periods to see if there are large changes in operational dynamics. We hope that this work can be further implemented into other large-scale hydrologic models to better represent reservoir dynamics and to evaluate their impact on hydrologic regimes and anthropogenic water availability.



*Code availability.*  PCR-GLOBWB 2 is freely available on the Utrecht University GitHub at https://github.com/UU-Hydro/PCR-GLOBWB_

model. All analysis and Validation codes are available upon request.

*Data availability.*  All the data in our analysis is available on YODA a Utrecht Universiteit wide data repository at https://public.yoda.uu.nl/
geo/UU01/F2UO5H.html and DOI:10.24416/UU01-F2UO5H

*Acknowledgements.*  All authors are grateful for the European Union's Horizon EUROPE Research and Innovation Programme under Grant
Agreement N° 101059264 (SOS-WATER -Water Resources System Safe Operating Space in a Changing Climate and Society) that financially

supported this work. We are also grateful for the larger development team at Utrecht University that supports the model development of PCR-
GLOBWB 2 and the two high performance computing clusters, without which this work would not be possible.

*Disclaimer.*

This project has received funding from European Union's Horizon EUROPE Research and Innovation Programme under
Grant Agreement N° 101059264 (SOS-WATER -Water Resources System Safe Operating Space in a Changing Climate and

Society). Views and opinions expressed are those of the author(s) only and do not necessarily reflect those of the European
Union or REA. Neither the European Union nor the granting authority can be held responsible for them.



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

## Appendix A

## A1

*Author contributions.* Jennie C. Steyaert, Marc Bierkens, and Niko Wanders discussed and developed the research trajectory. Jennie C. Steyaert performed all the analysis and took point on the writing, while Marc Bierkens and Niko Wanders provided valuable feedback in the writing and research phase. Edwin Sutanudjaja provided key support in developing the reservoir maps used in all of the analysis and supported the paper publication.





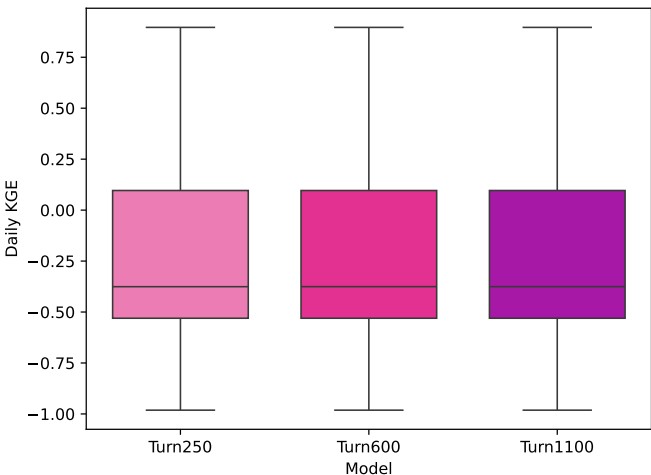

**Figure A1.** Shows a boxplot of the data derived operational scheme with the three main command areas: 250, 600 and 1100, in different colors.

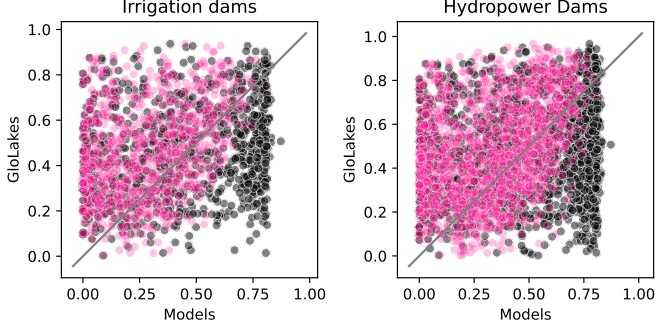

**Figure A2.** Depicts a scatter plot of the storage fractions in GloLakes vs the storage fractions in Turn250 and Baseline models. The left panel shows the results for dams with an irrigation main purpose and the right panel shows the dams with a hydropower main use.





*Competing interests.* Some authors are members of the editorial board of the HESS journal.