# Peer review of "Data derived reservoir operations simulated in a global hydrologic model"

_EGUsphere, 2024_

## Referee Comment (RC2)

I find this paper to largely be well written and its work addresses an important gap in the literature. There are some points I would like addressed and some room to improve clarity but overall I think it should be accepted after revisions.

**Some revisions to be considered:**
**1:** One topic I would like to see expanded upon is in relation to how certain sources of error in the input data and method may be influencing the final results. Specifically, the fact that storage values are what is used to constrain model training. To discuss why I think this is important I refer to.

Line 745-747 "Therefore, we suggest reservoir operation models rely primarily on validation of storage in place of validation solely on streamflow as the available streamflow observations are rarely close to the release point of the reservoir and therefore not as sensitive to reservoir operations compared to storage. "

While this may be true if the modelers in question are primarily concerned with the reproduction of storage, it seems to me that there are plenty of other sources of error that could make this untrue for other metrics. For one example, errors in the other fluxes of ET, precip, and storage lost to recharge could easily be introducing errors to the actual releases. Given the model is being trained to reproduce observed storage values, any errors in these fluxes will be baked into the final release. Even if this does not result in large streamflow differences downstream, if one cares more about release sizes than storage levels I could easily see streamflow data providing additional information.

I do want to acknowledge the authors have already provided analyses of underlying error, an example of which can be seen in lines 740-742:

"We observe that the RF extrapolation is accurately able to depict the flood and conservation curves and that the main source of uncertainty is the errors associated with the storage estimations from satellite altimetry."

What I think could be made a little more clear though is that the core role of storage in the training process, combined with the need to integrate many sources of data all with their own error, could mean there could easily be difficulties in reproducing less related metrics. And in fact, the improvements to storage related results do strongly outperform the improvements other metrics such as streamflow.

Hopefully I have not fundamentally misunderstood the paper when providing this comment. I do see that, for example, the authors validated against streamflow data as well and made sure to select only locations with measurements directly downstream. But my interpretation is that at the larger scale, estimates of storage are the only available data and thus the only available constraint.

I do not want to add a lot of work and whole new analysis to an already well written paper. I think simply a paragraph or two more directly acknowledging the challenges presented by these underlying errors with an illustrative example of one way that might play out, as well as qualifying some of the stronger statements in the introduction and conclusion making recommendations to modelers, would be sufficient.

**2:**
My second point is in regards to lines 250-251

"This spatial resolution is the optimal balance between computational demand and model performance and has been extensively validated and benchmarked"

I feel that this statement is too strong given the particular problem context. It may be true that this resolution has been extensively validated, but optimality is always a question of "optimal for what metric" and it is not clear to me that this previous work was looking at optimality under the same set of tradeoffs. For example, given that this method can be used to produce datasets, an outcome with a lot of potential downstream consumers, it may be optimal to use a lot more compute for even marginal gains in accuracy.

Additionally, this specific problem has characteristics that may mean a finer resolution is actually optimal. One in particular is discussed on the next several lines, and it is the fact that at this resolution some groups of reservoirs need to be considered as one reservoir because they share a grid cell. This nonlinearity presents what to me is a clear difference in trade offs from a simple performance/compute analysis.

To be clear, I am not suggesting this work should have been done at a different resolution, but this statement seems too broad.

**3:**
My third point is in regards to the analysis of the various components of the KGE in figure 5. Because all of the models performed very similarly on KGE overall, I am suspicious of reading too much into the size of the various components. Even an individual model with enough degrees of freedom may have parameters tunings that all produce the same overall KGE but with quite different component values. In this case, which component the model appears to perform better on will depend entirely on where you start your gradient descent.

So what is not clear to me is if the different methods have different component values because they are actually better suited to handle that component, or if they have different component values because they both have similar levels of ability to fit the data and have both settled at a somewhat arbitrary local minima that weights the components slightly differently.

**4:**
My fourth point is in regards to the command area analysis. The authors state the command area does not matter much, but I think this could be an artifact of the particular method.

Particularly, it is not clear to me that downstream demand plays a large role after the release curves have already been so constrained based on historical data. What I would like to see is a more clear description of through which equations the command area plays a role in determining the releases in the section performing the analysis. While the equations are described in the methods, it's a bit hard to sort out the answer to this specific question given the breadth of material being covered.

**5:** It feels like both the abstract and introduction could be shorter. For the introduction, some of the context being provided might be better suited to the methods section.

**6:** While I find the illustrative examples used to examine storage dynamics improvement useful, I think additional analysis needs to be done given the small sample they provide. Particularly, I would like to know how the examples compare to the average to know they have not been cherry picked.  Also, at least one of the selected examples should perform about average.

**Minor suggestions for improved clarity**
**7:** It took me a while to find that in the table 3 description it was specified that all RMSE values are in %/week. This made it very hard to interpret the results. I would suggest that these units be given wherever RMSE values are reported

Similarly, it seems the biases are being reported as percentages but that is not noted until many bias values have been reported. I similarly think the units should be specified at each location that biases are reported.

**8:** Figure 5.
Could the legend be made smaller to provide more room for wider graphs? They are narrow enough to be harder to interpret. Could also consider a 2x2 layout instead of 4x1. Also, it looks to me like the upper ylim was not set high enough for the KGE components and the distribution is being cut off at the top.

**9:** Lines 443-444: "Conversely, basins with a large amount of storage (Figure 2a) such as much of Central and South Eastern Asia, Central Africa, and Western Australia do not have a high degree of regulation"

I had to read this line a couple times to get it. I think changing to something like

"Conversely, **some** basins with a large amount of storage (Figure 2a) such as much of Central and South Eastern Asia, Central Africa, and Western Australia do not have a high degree of regulation, **which implies…**"

Would make what I believe to be the intended contrast to the previous lines more clear

**10:** Figure 3 y-axis just says %. It would be more immediately legible if it said % of what.

---

## Author Comment (AC1)

**Reviewer 1:**

Steyaert et al. present an approach to deriving reservoir operations for global scale hydrologic models. The study is of value to the water resources modeling community, but requires major revisions before being accepted for publication in HESS. Some elements of the method are not well justified, including the categorization of dams into "irrigation-like" and "hydro-like", as well as use of a release decision approach based on downstream demand aggregated across an arbitrary command area. Use of a random forest model to extrapolate curves is a nice idea but is not evaluated fully (i.e., using a cross-validation scheme) and appears quite ineffective based on the results shown. Although the level and depth of analysis conducted is impressive, the quality of results/figures is quite poor, and often confusing. The study can be simplified and reworked to deliver more clear and compelling results (with more impactful figures) on improvements offered by a data-derived storage scheme. The paper would also benefit from a significant reduction in number of words. The introduction is 13 paragraphs long and contains a lot of general detail. I encourage the authors to rewrite the introduction in a way that brings immediate focus to the problem area, most recent literature addressing that problem, and aims of the study. Three or four paragraphs will suffice. The abstract, currently almost 400 words, can be halved without loss of essential information.

**Thank you for your comment and for noting the importance of our work within the larger reservoir modelling community. We agree that the introduction and abstract can be shortened. We will shorten the abstract to 300 words or less and limit the total number of paragraphs in the introduction to seven that are focused on the following key points:**

- **The large number of dams and their impacts**
- **The multiple ways of modelling reservoirs and their current advantages and limitations**
- **How remotely sensed data can support the derivation of operational schemes.**
- **Our main research goals for this publication.**

**Comments:**

Title: Awkward repetition of "reservoir operations". Did you mean "A data derived workflow for simulating reservoir operations in a global hydrologic model" ? Also, this wording suggests that it is the *workflow* that is data derived, rather than the reservoir operation. So, did you actually mean something like "Data derived reservoir operations in a global hydrologic model" ?

**We agree that the repetition of reservoir operations leads to an awkward sentence and that the title sounds like the workflow is data derived. Therefore, we suggest to use your proposed title: "Data derived reservoir operations simulated in a global hydrologic model."**

Abstract L2. "most of the data was not openly accessible". I would suggest that this remains true. Specify the type of data.

We will include the following change: "Globally there are over 24,000 storage structures (e.g. dams and reservoirs) that contribute over 7,000km$^3$ of storage, yet until recently, most of these data was not openly accessible until recently."

L27. water supply reservoirs, flood control reservoirs, and hydropower dams are found in all climate types.
**Thank you for this comment, we will remove the regionality in this sentence. The sentence will read as follows: "With this loss of river connectivity comes a large amount of water storage (over 8,000,000 m$^3$ (Lehner et al., 2011) that provides water for a variety of purposes ranging from water supply and irrigation to hydropower and flood control."**

L187. Do you mean: "...to determine reservoir rule curves that specify seasonal flood and conservation pools..." ?
**Yes, this is a more clear and concise way to state what we are referring to. We suggest the following modification to the manuscript: "We input this weekly data into the STARFIT model developed by Turner et al. (2021) (Section 2.4.2) to determine reservoir rule curves that specify seasonal flood and conservation pools. After obtaining seasonal flood and conservation pools for 1752 reservoir, ..."**

L205. Not clear what is meant by "yearly maps of static reservoir characteristics".
**This refers to the reservoir characteristics used as inputs for PCRGLOBWB2. These maps are used to determine 1) where reservoirs exist and 2) the necessary hydrologic characteristics (outlet points, storage capacity, reservoir id, and surface area) that are used to calculate storage within the model. This input is given to the model as they do not change frequently, however, this also means that new reservoirs will always appears on January 1$^{st}$ and will only contribute to the river management from that day until they are removed (if this occurs during the simulation period). To make this clearer, we propose the following change: "From this updated table, we created annual maps of static reservoir characteristics (e.g. outlet points, storage capacity, reservoir id, and surface area), which are used as inputs to model reservoir releases and to distinguish between two operational policies *hydropower-like* and *irrigation-like.*"**

Also, since L180 I have been reading and wondering the motivation and reasoning behind these two categories ("hydropower-like" and "irrigation-like"). Please try to clarify the role of this categorization early in the study.
**Thank you for your comment. We agree that explaining this classification earlier in the manuscript is useful. We will add a description at line 180 to clarify what these two groupings are. The updated sentence reads as follows: "Using these operational bounds, we derive two main reservoir models *for irrigation-like* (dams that are focused on meeting downstream demand) and *hydropower-like* dams (dams that are focused on holding storage stable). We will also edit the following description at line 205 as follows: "We separated our operations into these two categories as Steyaert and Condon and Salwey et al. noted differences in operational patterns between storage reservoirs (noted as irrigation and water**

supply main uses) and non-storage reservoirs (such as hydropower, navigation and flood control uses)."

L250. Please add further detail here on whether any efforts were made to ensure reservoirs were placed on correct streams. From what I read, it seems the lat/lon of the reservoirs are snapped to the PCR-GLOBWB grid then assigned that grid cell.

**To correctly match the dams, we calculated the closest grid cells in PCR-GLOBWB 2 to the latitude and longitude reported in GeoDAR and the catchment areas of each grid cell in PCR-GLOBWB 2. We then minimized the eucludian distance between the grid cell and the location of the dam and the difference between reported catchment area and the catchment area on the PCR-GLOBWB 2 domain. This ensures that the GeoDAR dam is mapped to the correct stream and that the entire reservoir sits within a single catchment. In some cases, this information is missing from GeoDAR and we therefore spatially snapped the reservoirs to the nearest latitude and longitude point on the river network. While this could lead to inaccuracies, the 5 minute spatial resolution (approx. 10km) typically contains the largest rivers in the network, so it is not likely that we are mapping large dams to very small rivers. To clarify this in the manuscript we will include the following description on line 250: "We then ensured that the mapped location based on the latitudes and longitudes from GeoDAR also aligned with other reservoir characteristics such as catchment area. We compared the catchment areas reported in GRanD, iCOLD and GeoDAR to the calculated catchment area at the dam location calculated from the PCR-GLOBWB 2 DEM. For each potential location, we minimized the difference in catchment area and the distance to the reported latitude and longitude of the dam."**

L270. Ok—here I am now realizing that irrigation-like and hydropower-like categories are used to inform releases, with the starfit approach solely defining storage curves. Doesn't this mean the operations are not full data-driven but rather half data driven (storage curves) and half "generic" (release policy based on command area demand and reservoir purpose)?

**Unlike Turner et al., 2022, we were unable to gather enough reservoir data to fully derive the reservoir releases using a purely data derived method as in most cases data for reservoir releases is missing. Therefore, we opted to only derive reservoir storage bounds using static reservoir characteristics described in Section 2.5. These reservoir storage bounds denote the active area within which reservoir release is defined by the equations in Section 2.4. We, therefore, use the two main groupings, *irrigation-like* and *hydropower-like*, to steer the release equations. Both of these groupings take into downstream demand that has been aggregated along the downstream areas of 250, 600, or 1100. We opted to use this instead of a generic scheme as Steyaert and Condon ( 2024) noted that hydropower and navigation dominate regions in the United States have a more stable reservoir storage compared to regions dominated by irrigation and water supply uses.**

**In order to clarify this, we plan to update Line 270 – 272 to include the following: "As our analysis is done globally, we use data from the 1752 dams in data from 1752**

dams in GloLAKES (Hou et al., 2024) and derive the operational bounds for the STARFIT using a combination of observations and machine learning. To compliment these operational bounds, we employ two main sets of equations based on two main groupings of reservoir main purposes: *irrigation-like* and *hydropower-like* (Section 2.4.4 and Section 2.4.3). We use these two groupings to denote how releases change based on the level of storage. In *irrigation-like* dams, the goal is to meet downstream demand and therefore the equations in Section 2.4.1 prioritize this goals by meeting all downstream demand when reservoir storage sits between the data derived operational bounds and proportionally less when storage sits between the conservation bound and 10% of the maximum storage capacity of the reservoir. For *hydropower-like* dams, the goal is to hold storage as stable as possible. Therefore, the equations in Section 2.4.2 prioritize meeting downstream demand when the storage in the hydropower-like reservoir sits between the data derived operational bounds. However, if meeting this downstream demand causes the reservoir storage to drop below the conservation bound, then the reservoir can only meet a portion of demand to allow storage to stay in the active zone (zone between the operational bounds). For both types of reservoirs, we employ an additional flood release and account for environmental flow requirements as described in Gleeson and Wada (2013)."

L313 – missing reference to equation 5.
We will add the requested reference. Please see the following edits: "To do this, these daily storage, release and inflow values are aggregated into weekly time series and a combination of sine and cosine curves (described by equation 5 below) are fit to the upper and lower percentiles of each time series."

L325-330. I would be very unsure about labels of water supply / irrigation vs hydro etc within GranD leading to a neat splitting of dams respectively operated for downstream demand versus maintaining high storage levels. Apart from the issue of inaccurate reservoir purposes in the available global datasets, one rarely finds such simple distinctions in reality. Are you able to show that two categories of operations actually exist, e.g., by comparing the starfit curves for irrigation-like versus hydropower-like dams in the set of 1752 observed dams? I would be surprised if you find a clear distinction. If this is the case, I don't see strong justification for the splittling—which in a way complicates the study.
We kindly thanks the reviewer for this comment. We do agree that there may be inaccuracies in the main uses in GranD. This said, GRanD is still the leading dataset for determining reservoir main uses. As shown in Figure 3 in the manuscript, there are differences in the two main categories of reservoirs we used, however, we agree that analysis of the StarFIT curves for differences in the operations is useful. In the following figure (Figure 1), we plot the average, maximum and minimum value of the derived STARFIT curves for the *irrigation-like* (blue) and *hydropower-like* (red) dams for both the flood (Figure 1, top row) and the conservation (Figure 1, bottom row) bounds. While the average and maximum flood and the maximum conservation values do not differ much between the dams, we do see large differences in the average conservation and the minimum flood and conservation curves which could be a result of the differing operations at the lower end of storage. Specifically, the

flood minimum peaks in irrigation type dams in the spring and summer months to potentially support downstream demands in more drier periods, while the *hydropower-like* dams have lower flood minimum values. The conservation curves experience the most changes in part due to the *hydropower-like* dams holding storage much higher across the year while the *irrigation-like* dams are meeting downstream demand in the autumn months. For the minimum conservation values, the *irrigation-like* dams have higher storage fractions compared to the *hydropower-like*. Due to the differences in the seasonality of the lower bounds for the flood curve and the differences in the conservation curves, we still think that the distinction in operational schemes is useful.  We will include this figure in the supplemental as well as the above description.

[Figure]

*Figure 1: Depicts the average, maximum and minimum flood (top row) and conservation (bottom row) curves that are used in PCR-GLOBWB2*

L335. It's unclear to me what the command area offers. The storage curves can guide the release without a downstream demand. Were any tests performed to evaluate whether this downstream demand actually improves on accuracy?
**The storage curves are able to guide a release without a downstream area, however, we wanted to include the downstream water demand dependencies. In addition, we wanted to test the sensitivity of streamflow to difference in these three command areas typically described in the literature. We did not solely isolate the downstream command areas in our analysis; however, we do show in our results that the curves separated by reservoir use and using a command area do provide a more accurate representation of reservoir storage globally (Figure 6 and Figure 7 in the submitted manuscript). We acknowledge that it would be useful to**

perform a more comprehensive test to see if differences in the command area do contribute to changes in our operational scheme. To do this, we re-ran our model set up for the Mississippi basin and set the downstream command area to 0 which, when multiplied by the downstream demand, removes the demand. We then evaluated the daily streamflow KGE values (Figure 2) to observe the differences between the previous model runs and the model run without the command area. We therefore suggest adding the following figure showing the CDF of the daily KGE values plotted for the Baseline (Figure 2, black), vanBeekGeo (Figure 2, grey), Turn250 (Figure 2, pink) and the Turner operations without a command area (Figure 2, purple) for the Mississippi Basin to the supplemental. From Figure 2, we do see that the addition of the command area and accounting for downstream demand does improve streamflow dynamics when using the two reservoir groupings (*hydropower-like* and *irrigation-like* dams).

To clarify this in the text, we will add the following: "If during this process, another dam intersects the river network before the full command area is created, we assume that this is the maximum distance that is served by the upstream reservoir. This command area is used to aggregate the total downstream demand that could be met by the reservoir. We use this aggregated downstream demand in both the *hydropower-like* and *irrigation-like* dams as both dam types can meet the downstream demand when storage sits between the data derived operational bounds. We found that while our model was not sensitive to the downstream area (Supplementary A1), we did observe that the addition of a command area increased our model performance." We will include a reference to Figure 2 in the text and the figure in the supplementary.

[Figure]

*Figure 2: Cumulative distribution function of the daily KGE values for the Mississippi Basin. The colored lines depict the three original scenarios: Baseline (black), vanBeekGeo (grey) and Turn250 (pink), with an additional line (purpl and labelled No Command Area) that depicts the streamflow distribution if there were no command areas.*

L342. How are surface water abstractions considered? Is this based on demand within the same grid cell as the reservoir?

**In PCR-GLOBWB 2, surface water is abstracted from the closest water body or river to the grid cell within a 100km radius that has demand. We updated this scheme to only abstract water from the reservoir if it is in our *irrigation-like* category and if the abstraction would not cause the dam to drop below the conservation level. We will include the following lines to explain this further: "In PCR-GLOBWB 2, surface water is abstracted from the river or lake cell closest to the cell with a demand. We changed this scheme to limit surface water abstraction to *irrigation- like* dams, and only to the extent that the abstracted volume of water would not drop the reservoir storage below the conservation curve."**

Equation 6. Maybe I missed this, but how is Env defined? Also, how is the flood release defined? Is this just spill required to draw the reservoir back to the active zone?

**Thank you for noticing this. We left off the description of Env. We will edit the section as follows: "Lastly, we implement a piecewise function for releases based on the current reservoir storage (Sc) where Rf is the flood release, Env is the environmental flow requirement defined in PCR-GLOBWB 2 as 10% of the naturalized flow (Gleeson & Wada, 2013). Ri and Rh are the irrigation and hydropower releases in the active zone and are described in by equation 9 in Section 2.4.4 and by equation 8 in Section 2.4.3 respectively."**

**Yes, this flood release is the release needed to draw the reservoir back to the active zone.**

L350. Unclear what is being done here. Are you creating an active zone per dam type and country? Why? I thought the random forest provides full parameterization for each dam.

**We are not providing an active zone per dam type and country, but an active zone per dam based on a random forest algorithm, where type of use, socioeconomic and climatic variables are used as features (predictors). We then use a set of equations to simulate release based on downstream demand. While it is ensured that the reservoir storage stays within the active zone defined by the random forest algorithm (Equations 6-9). We define two main categories of equations for *irrigation-like* and *hydropower-like* reservoirs to simulate the different dynamics within each. For *hydropower-like* reservoirs, the equations assume that the operator is attempting to keep reservoir storage in the active zone as much as possible and there are no releases if the reservoir is below the active zone. For *irrigation-like* reservoirs, the goal is to meet all the downstream demand within the command area.**

**To clarify our workflow, we have added the following paragraph explaining the differences at Line 270. "To compliment these operational bounds, we employ two main sets of equations based on two main groupings of reservoir main purposes:**

*irrigation-like* and *hydropower-like* (Section 2.4.4 and Section 2.4.3). We use these two groupings to denote how releases change based on the level of storage. In *irrigation-like* dams, the goal is to meet as much downstream demand and therefore the equations in Section 2.4.1 prioritize meeting downstream demand with more downstream demand met when the storage sits between the data derived operational bounds and proportionally less when the storage sits between the conservation bound and 10% of the maximum storage capacity of the reservoir. For *hydropower-like* dams, the goal is to hold storage as stable as possible. Therefore, the equations in Section 2.4.2 prioritize meeting downstream demand when the storage in the *hydropower-like* reservoir sits between the data derived operational bounds. However, if meeting this downstream demand causes the reservoir storage to drop below the conservation bound, then the reservoir can only meet a portion of demand to allow storage to stay in the active zone (zone between the operational bounds). For both types of reservoirs, we employ an additional flood release and also account for environmental flow requirements as described in Gleeson and Wada (2013)."

We also suggest combining Sections 2.4.2, 2.4.3 and 2.4.4 to one section titled "Data Driven Reservoir Operations-STARFIT," with three subsections defined as 1) Operational curves by STARFIT, 2) operations for *hydropower-like* dams, and 3) operations for *irrigation-like* dams. We also plan to add the above text to the beginning of Section 2.4.2. Lastly, we also suggest adding the follow text to line 350: "We use these operational bounds to denote the active zone and therefore the release factor (Equation 4) for the hydropower dam. We opted for different hydropower and irrigation operations as the main goal of each type of reservoir is slightly different. For example, a hydropower dam in Switzerland could have slightly different operational bounds than a hydropower dam in Vietnam, however the main purpose: hold enough water to support electricity generation, would be the same."

L381. After validating the model and demonstrating effectiveness with the 25% out validation, why not re-train with all 1,752 structures before extrapolating? Also, given the importance of the random forest to the overall framework, I strongly suggest the authors pursue a k-fold cross validation scheme rather than single training and test samples.
**Thank you for the comment. We did retrain all the structures as well as the 1,752 before extrapolating. We will update line 381 to read: "The obtained RF was then used to extrapolate the 10 parameters to all 24,000 structures." We also think a k-fold cross validation could be useful to validation. We ran a test with the 1752 dams with the same 75% training and 25% testing split as the single RF method, meaning we put 75% of the data through the k-fold validation and kept 25% out to validate and test our method. The k-fold cross validation splits the data into 10 equal portions. We then created a composite score of the MAE and MSE to determine the overall best model from the k-fold using the 25% of the data we left out for validation. For all 10 models we received the following results for the mean**

squared error, mean absolute error and the r squared comparing the random forest models predictions to the Turner values.

| Model | k-fold with cv = 10 | Best K-fold cross validation model | Single RF method |
|---|---|---|---|
| MSE | 359.77 (stdev = 47.13) | 291.15 | 288.39 |
| MAE | 12.96 (stdev = 0.83) | 11.74 | 11.65 |

From these results, we see that the current random forest setup has a lower MSE and MAE values suggesting the single RF method is performing well. The k-fold cross validation does show us that there is some sensitivity to our testing and training dataset due to the standard deviations of the MSE and MAE. Our initial setup performs slightly better when looking at the MSE and MAE as the errors associated with the single RF methodology are lower. Therefore, we think it is justified to use the full dataset for the RF, however we already noted in the discussion that the extrapolation of parameter values is an area of uncertainty that could be further reduced by using different techniques or more data and we will provide the above table depicting the results of the cross validation in the appendix.

The addition to Line 588 will read as follows: "Additionally, we may find that by using a different validation scheme, our operational curves may also change as our random forest is sensitive to the input data."

L385. How many reservoirs end up being constrained to these bounds? Also, it's not clear what is meant by flood peak here. Do you mean upper bound of active storage? Table 2. Here would be very interesting to see a version that drops the command area and demand parameters (as well as hydro/irrigation split) entirely. I can't see a strong justification for the demand-based release or the command area (or the hydro / irrigation split for that matter). A simple way to test this would be to take the mid-point of the active zone (i.e. assume just one curve to target) and operate toward that at all times (giving you a very simple release function).

**Thank you for your comment. We decided to implement a simple rule curve for the Mississippi Basin that accounts for the downstream demand (the green line in Figure 3 and Figure 4). This simplified operational policy still accounts for downstream demand according to the 250km distance and can meet this demand and surface water abstractions if storage is within the active zone (definied as the area between the flood and conservation curves) and includes environmental flow and flood releases. We ran the model for the Mississippi basin without the command area (by setting the downstream demand to 0) but including the two operational schemes (purple line). In analyzing the longterm monthly storage for the simple rule curve we observe that we hold less water on average, but the seasonal dynamics are similar to the other models ( Figure 3). This suggests that the biggest difference is the overall storage fraction levels and in fact this simplified rule curve decreases the overall water availability in the Mississippi region.**

We then computed the daily KGE values for these two models as well as the Turn250, Baseline and vanBeekGeo against the streamflow observations in GRDC. While the addition of the command area slighlty improves the model (Figure 4, purple vs pink lines), we do see large improvements in using two different operational schemes (green vs pink lines in Figure 4). This suggests that creating two different release rules for irrigation-like and hydropower-like dams enhances model performances compared to a single simplified scheme. We also saw there are operational differences in the average conservation curves and the flood and conservation minimum curves when looking at the two typologies we defined (Figure 1 above and copied below).  This, in conjunction with Steyaert et al., 2024 and Salwey et al., 2023 noting that there are differences in irrigation, water supply and hydropower dams, further supports our conclusion that having two main types of reservoir operations better represents the observed dynamics. We plan to include these two figures (Figure 3 and Figure 4) as well as the above explanation in the supplementary.

[Figure]

*Figure 3: Longterm storage fraction of the different models in our analysis (Turn250 in pink, Baseline in black, and vanBeekGeo in grey) as well as the simple rule curve (green).*

[Figure]

Figure 4: Depicts the cumulative distribution function of the daily streamflow KGE values for the original model scenarios: Baseline (black), vanBeekGeo (grey) and Turn250 (pink) and the simplified rule curve (green).

[Figure]

Figure 5: Depicts the average, maximum and minimum flood (top row) and conservation (bottom row) curves that are used in PCR-GLOBWB2

L503. Above you state that Clinton dam has a hydropower main purpose.
**You are correct. We initially used the Clinton dam here, but changed the dams in the final version of the code but did not account for these changes in our mansucript. Figure 3 in the manuscript shows Butt Valley dam in California for hydropower use and Figure 4 in the manuscript shows Clinton Lake Dam which has a water supply main use and Koelnbrein dam which is a hydropower main use. We will correct the manuscript accordingly.**

Figure 4. Is this average monthly discharge over a number of years, or are you showing a single year's output?

**We are showing the longterm monthly average discharge over the model period. We will update the caption accordingly**

L588 – this is an inadequate way to evaluate storage dynamics improvement. You have observation and results. Compute NSE / RMSE / KGE or similar for each dam (sim vs obs) and show the difference across a distribution (perhaps splitting by continent or large basin).
**Thank you for this comment. We agree that adding a plot showing the improvement by calculating the KGE, NSE or RMSE between our observations and simulations would be a useful addition. Instead of including all three, we opted to show the KGE and RMSE between the modelled values and the observations as global CDFs (Figure 6). The KGE plot shows that the Turn250 model has relatively more negative KGE values, however, these negative performances are typically in wetter periods where PCR-GLOBWB 2is already underestimating streamflow. This model also has larger KGE values. As for the RMSE we do see that the Turn250 has more values closer to 0 suggesting the Turn250 model is more aligned with the observations. We also opted to plot the KGE components (Figure 7). The alpha and R components show slight improvements in modelled storage with the Turn250 operations, while the beta shows that the Turn250 has more bias, which is most likely occuring in the wetter periods. To supplement this, we will include the above description and the following figures to the supplementary.**

[Figure]

*Figure 6: Cumulative distribution plots of the monthly storage KGE and monthly storage RMSE for the Baseline (black), vanBeekGeo (grey) and the Turn250 (pink) models*

[Figure]

*Figure 7: Cumulative distribution plots of the storage KGE components: alpha, beta and the cross correlation (CC), for the three models in our analysis: Baseline (black), vanBeekGeo ( grey) and Turn250 (pink).*

Figure 7. It's not clear why the data-derived storage curves result in a different seasonal storage pattern than GloLAKES for North America. Aren't the curves based on GloLAKES data?

**Yes, the curves are based on the data in GloLAKES and therefore should align, however, the number of US dams in GloLAKES (1752 with 543 or 31% in the US plotted in red in Figure 9) differs from the total number of dams (over 20,000 with 8214 or 40% in the US plotted in blue in Figure 10). Additionally, the random forest algorithm looks for similarities and differences across all the dams in the training set. This training set (75% of all the data) is chosen randomly and, while it includes dams from the US, we make sure to choose multiple regions. Therefore, this could account for the regional differences in our storage patterns compared to the GloLAKES observations. When plotting the monthly KGE and monthly RMSE (Figure**

**8) for each of the models, we do see that the RMSE in the United States are much higher and the KGE is slightly worse. This suggests that the issue in performance is perhaps due to the underlying model dynamics in PCR-GLOBWB 2 as well as the inclusion of other regions in the training dataset to create the Random Forest algorithm. We will include the following figures as well as the above description in the supplement.**

[Figure]

*Figure 8: Cumulative distribution plots of the monthly storage KGE and monthly storage RMSE for the Baseline (black), vanBeekGeo (grey) and the Turn250 (pink) models across the United States.*

[Figure]

*Figure 9: Map of the point locations of the Glolakes observations used to train our random forest algorithm and validate our analysis.*

[Figure]

*Figure 10: Point location map of all the dam locations in GeoDAR that are included in our analysis.*

---

## Author Comment (AC2)

**Reviewer 2:**

I find this paper to largely be well written and its work addresses an important gap in the literature. There are some points I would like addressed and some room to improve clarity but overall I think it should be accepted after revisions.

**Thank you for your kind words and comments.**

**Some revisions to be considered:**
**1:** One topic I would like to see expanded upon is in relation to how certain sources of error in the input data and method may be influencing the final results. Specifically, the fact that storage values are what is used to constrain model training. To discuss why I think this is important I refer to.

Line 745-747 "Therefore, we suggest reservoir operation models rely primarily on validation of storage in place of validation solely on streamflow as the available streamflow observations are rarely close to the release point of the reservoir and therefore not as sensitive to reservoir operations compared to storage. "

While this may be true if the modelers in question are primarily concerned with the reproduction of storage, it seems to me that there are plenty of other sources of error that could make this untrue for other metrics. For one example, errors in the other fluxes of ET, precip, and storage lost to recharge could easily be introducing errors to the actual releases. Given the model is being trained to reproduce observed storage values, any errors in these fluxes will be baked into the final release. Even if this does not result in large streamflow differences downstream, if one cares more about release sizes than storage levels I could easily see streamflow data providing additional information.

**Thank you for your comments. We do agree that there could be issues that are propogated from other soures of error. We suggest changing Lines 745 – 747 to read as follows: "emphasize model validation on reservoir storage in addition to validation based on streamflow," in place of validation solely on streamflow as available streamflow observations are rarely close to the release point of the reservoir and therefore not as sensitive to reservoir operations compared to storage.**

I do want to acknowledge the authors have already provided analyses of underlying error, an example of which can be seen in lines 740-742:

"We observe that the RF extrapolation is accurately able to depict the flood and conservation curves and that the main source of uncertainty is the errors associated with the storage estimations from satellite altimetry."

What I think could be made a little more clear though is that the core role of storage in the training process, combined with the need to integrate many sources of data all with their own error, could mean there could easily be difficulties in reproducing less related metrics. And in fact, the improvements to storage related results do strongly outperform the improvements other metrics such as streamflow.

Hopefully I have not fundamentally misunderstood the paper when providing this comment. I do see that, for example, the authors validated against streamflow data as well and made sure to select only locations with measurements directly downstream. But my interpretation is that at the larger scale, estimates of storage are the only available data and thus the only available constraint.

I do not want to add a lot of work and whole new analysis to an already well written paper. I think simply a paragraph or two more directly acknowledging the challenges presented by these underlying errors with an illustrative example of one way that might play out, as well as qualifying some of the stronger statements in the introduction and conclusion making recommendations to modelers, would be sufficient.

**Thank you for your comprehensive comment. We do agree that adding a section on the propogation of errors would enhance the paper. We suggest this paragraph falls under Section 4.2 and contains the following:**

- **Sources of potential error using PCR-GLOBWB 2 inputs, which potentially cancel out the associated errors in the PCR-GLOBWB 2 model**
- **Sources of error in using remotely sensed data**
- **Sources of error in only looking at storage as a validation method.**
- **The impact these errors have on the storage and release of the reservoir**

**We will include the following paragraph regarding this error propogation:**

**"Apart from errors accruing from above assumptions, the accuracy of our results is also limited by the errors that are propogated through our workflow. Specifically, PCR-GLOWBWB 2 underestimates the flashiness of streamflow regimes. It is also less accurate in specific regions such as the Niger, the Rocky Mountains and portions of continental Eastern Europe due to errors in the snow dynamics, estimation of the groundwater responses and data limitations (Sutanudjaja et al., 2018). Additionally, the estimation of the operational STARFIT rules from the remotely sensed storage data of GloLAKES is limited by the revisit time of satellites, the influence of cloud cover and atmospheric interference as well as the statistical models that back calculate storage that are limited by the digitial elevation model resolution (Hou et al., 2024; Chen et al., 2022). As storage is typically not a measured value and, even in the case of in-situ observed water levels observations, is back calculated from storage/area or storage/elevation relationships, validation primarily on storage alone is inherent to uncertainty. Primarily, these limitations can affect the actual storage value as they rely on storage elevation charts that are only periodically updated (Steyaert et al., 2022) While the single errors are propogated through our system, the results of the independent validation with ResOpsUS (Figure 6 in the manuscript) and GloLAKES (Figure 6 and Figure 7 in the manuscript) show improved performance for storage values in PCR-GLOBWB 2 and suggest similar improvements for other global hydrologic models with the caveat that errors may propagate through the modelling system."**

**2:**

My second point is in regards to lines 250-251

"This spatial resolution is the optimal balance between computational demand and model performance and has been extensively validated and benchmarked"

I feel that this statement is too strong given the particular problem context. It may be true that this resolution has been extensively validated, but optimality is always a question of "optimal for what metric" and it is not clear to me that this previous work was looking at optimality under the same set of tradeoffs. For example, given that this method can be used to produce datasets, an outcome with a lot of potential downstream consumers, it may be optimal to use a lot more compute for even marginal gains in accuracy.

Additionally, this specific problem has characteristics that may mean a finer resolution is actually optimal. One in particular is discussed on the next several lines, and it is the fact that at this resolution some groups of reservoirs need to be considered as one reservoir because they share a grid cell. This nonlinearity presents what to me is a clear difference in trade offs from a simple performance/compute analysis.

To be clear, I am not suggesting this work should have been done at a different resolution, but this statement seems too broad.

**This is a really good point. We intially meant this statement to refer to the computational time for running our model on the global scale. By moving to a higher resolution of the PCRGLOBWB 2 model (such as the 30 second resolution), we introduce more potential errors in land cover type and snow dynamics that further complicate the results due to increased evapotranspiration from crop types and lack of lateral transport for snow (van Jaarsveld et al 2025). Additionally, running the PCRGLOBWB 2 model globally on the 30 second resolution takes 401 computational hours accoding to van Jaarsveld et al., 2024. Ultimately, we agree that this is a broad statement and suggest the following change: "We opt for the 5 minute resolution in order to capitalize on the extensive validation and benchmarking done by Sutandujaja et al., 2018 and to limit excessive calculation times that occur at higher resolutions (van Jaarsveld et al., 2025)."**

**3:**

My third point is in regards to the analysis of the various components of the KGE in figure 5. Because all of the models performed very similarly on KGE overall, I am suspicious of reading too much into the size of the various components. Even an individual model with enough degrees of freedom may have parameters tunings that all produce the same overall KGE but with quite different component values. In this case, which component the model appears to perform better on will depend entirely on where you start your gradient descent.

So what is not clear to me is if the different methods have different component values because they are actually better suited to handle that component, or if they have different component values because they both have similar levels of ability to fit the data and have both settled at a somewhat arbitrary local minima that weights the components slightly differently.

**This is a really good point. To expand on this point, we created scatters plots of the different KGE components between the Turn250 and vanBeekGeo models (Figure 1). While the scatter for the R component makes this component appear to be the most important, we find that both the R and beta components have almost equal values above and below the 1:1 line suggesting that these two components are muting the KGE differences. Comparatively, alpha has 1196 points above the 1:1 line and 779 points below the 1:1 line which suggests that alpha is the most sensitive to the operational changes and contributes the most to the KGE changes (1210 above the 1:1 line and 1158 below the 1:1 line). To show this in the analysis, we propose to add this figure and analysis to the supplementary.**

[Figure]

*Figure 1: Scatter plots of the streamflow KGE components between each model and observations. We plot the KGE components ( alpha, beta, and R) for the Turn250 model on the y axis and the KGE components for the vanBeekGeo model on the x axis. The dashed red line is the one to one line.*

**4:**

My fourth point is in regards to the command area analysis. The authors state the command area does not matter much, but I think this could be an artifact of the particular method.

Particularly, it is not clear to me that downstream demand plays a large role after the release curves have already been so constrained based on historical data. What I would like to see is a more clear description of through which equations the command area plays a role in determining the releases in the section performing the analysis. While the equations are described in the methods, it's a bit hard to sort out the answer to this specific question given the breadth of material being covered.

**Thank you for your comments. The command areas are taken into account in equations 7 and 9. We calculate the command area as the downstream region that the reservoir could supply water to. Therefore, D in equations 7 and 9 refers to the maximum downstream demand that is aggregated over the specified command area per model (i.e. 250, 600, and 1100). We suggest adding the following to clarify this on line 356: "where D refers to the maximum demand aggregated at the specified downstream area (250, 650, 1100)."**

**5:** It feels like both the abstract and introduction could be shorter. For the introduction, some of the context being provided might be better suited to the methods section.

**We agree that shortening the introduction and the abstract would be useful and will shorten the abstract to less than 300 words. We will also shorten the introduction from 13 paragraphs to 7. We do think some of the context is quite lengthy and we can still cover the main components in a simplier fashion.**

**6:** While I find the illustrative examples used to examine storage dynamics improvement useful, I think additional analysis needs to be done given the small sample they provide. Particularly, I would like to know how the examples compare to the average to know they have not been cherry picked. Also, at least one of the selected examples should perform about average.

**Thank you for your comment. We thought the single point location was a nice way to illustrate the potentiall differences in operational dynamics and their impacts. We agree that a point location does not tell the full story. To better tell this story, we have opted to include the climatology of the storage fraction and the storage integral to show the average changes between the different model scenarios. From this figure, we observe on average that the storage fractions in Figure 4 in the manuscript align with the general trends we see in the average storage fraction climatology (Figure 2). That said, the average storage values are lower in modelled values in Figure 4 in the manuscript compared to all the dams in our analysis. To compliment this qualitative analysis, we also calculated the correlation and the KGE for the three models between the longterm monthly storage of all the dams and the Clinton and Koelnbrein dams (below table). We do observe that the Koelnbrein dam has high correlations and slightly positive KGE values that suggest that this dam is fairly representative of the dynamics we observe when taking the average of all the dams in the longterm storage. The Clinton dam, however, has a varied performance depending on the model suggesting that this dam has different dynamics than the longterm monthly storage values.**

| Model | Clinton KGE (storage) | Clinton R (storage) | Clinton KGE (storage integral) | Clinton R (storage integral) |
|---|---|---|---|---|
| **Baseline** | 0.068 | 0.05 | -7.95e17 | -0.88 |
| **vanBeekGeo** | -0.45 | 0.43 | -3.37e18 | -0.69 |
| **Turn250** | -0.219 | -0.032 | -2.01e18 | -0.911 |

| Model | Koelnbrein KGE (storage) | Koelnbrein R (storage) | Koelnbrein KGE (storage integral) | Koelnbrein R (storage integral) |
|---|---|---|---|---|
| **Baseline** | 0.13 | 0.96 | -4.47e17 | 0.41 |
| **vanBeekGeo** | 0.25 | 0.85 | -5.62e17 | 0.65 |
| **Turn250** | 0.13 | 0.85 | -7.56e17 | 0.56 |

**When looking at the storage integral average climatology, we see varied dynamics in the summer months that align with the average of the two examples in Figure 4 in the manuscript. However, the KGE and correlation values show that the Clinton dam is**

not well represented by the longterm average plots and the Koelnbrein dam on the other hand is representative of the average dynamics. Therefore, the two examples shown in Figure 4 in the manuscript both show an example of a dam that aligns with the expected average values (Koelnbrein dam) and an example of a dam that is not indicative of the average trends (Clinton dam). We plan to include this figure (Figure 2) and the two tables in the supplementary and keep figure 4 in the manuscript as is. We will also include plots of the longterm monthly discharge at major basin outlets so the regional differences can be better seen and add this plot to the supplementary.

[Figure]

*Figure 2: Plots of the longterm storage fraction (left) and the longterm storage integral (right) for the three models: Baseline (black), vanBeekGeo (grey) and Turn250 (pink).*

**Minor suggestions for improved clarity**
**7:** It took me a while to find that in the table 3 description it was specified that all RMSE values are in %/week. This made it very hard to interpret the results. I would suggest that these units be given wherever RMSE values are reported

Similarly, it seems the biases are being reported as percentages but that is not noted until many bias values have been reported. I similarly think the units should be specified at each location that biases are reported.

**Thank you for your comment. We will update the table to include the units. We also will go through the manuscript and make sure the units are stated when we first mention the metric.**

**8:** Figure 5.
Could the legend be made smaller to provide more room for wider graphs? They are narrow enough to be harder to interpret. Could also consider a 2x2 layout instead of 4x1. Also, it

looks to me like the upper ylim was not set high enough for the KGE components and the distribution is being cut off at the top.

**We agree that a 2x2 layout would align better with this figure. We specifically set the ylim in order to see the small differences in the distribution as the majority of values were at 0 for alpha, beta and and R. We will update the original figure to include the 2x2 panel (Figure 4) and will put the plot without the zoom (Figure 3) in the supplementary.**

[Figure]

Figure 3: Show the KGE, and KGE components of the three models we used in our analysis for Figure 5 in the manuscript without any zoom.

[Figure]

Figure 4 Show the KGE, and KGE components of the three models we used in our analysis for Figure 5 in the manuscript as a 2x2 panel plot.

**9:** Lines 443-444: "Conversely, basins with a large amount of storage (Figure 2a) such as much of Central and South Eastern Asia, Central Africa, and Western Australia do not have a high degree of regulation"

I had to read this line a couple times to get it. I think changing to something like

"Conversely, **some** basins with a large amount of storage (Figure 2a) such as much of Central and South Eastern Asia, Central Africa, and Western Australia do not have a high degree of regulation, **which implies...**"

**Thank you for the comment. We think this is a really nice change and will ammend the manuscript as follows: "Conversely, some basins with a large amount of storage (Figure 2a) such as much of Central and Southeastern Asia, Central Africa, and Western Australia do not have a high degree of regulation, which implies that there is not a direct relationship between total storage and a high degree of regulation (Figure 2b)"**

Would make what I believe to be the intended contrast to the previous lines more clear **10:** Figure 3 y-axis just says %. It would be more immediately legible if it said % of what.

**We agree that this could be more clear. We will change the axis to include Storage Percent (%).**

---

## Author Response (AR2)

**Reviewer 1:**

The authors have dealt with most of my prior comments comprehensively. My final recommendation is on the interpretation of the new result relating to the command area (shown in appendix 4). This limited analysis (only daily KGE) over the one basin (Mississippi) shows that the command area adds only very marginal value, and gives the reader no strong sense of the value of command area in general. Instead of further trying to defend the use of a command area without strong evidence of its value, the authors may instead reflect on this result and ask whether downstream demand areas are really worth adopting in global hydrologic models with reservoirs. Better yet (and this is something I strongly encourage), run the global simulation (all basins) without the demand area in place so there is a definitive answer on whether this feature is worthwhile. If the Mississippi result is reflected elsewhere, the authors can make an impactful statement on the real drivers of reservoir operations globally--i.e., set by storage curves and generally uninfluenced by downstream demand, at least if demarked by an arbitrary command boundary. This would be useful information to inform future efforts and new approaches to representing dams at global scale, as it would suggest efforts ought to be placed on better understanding storage curves and operations with respect to storage conditions, which are increasingly available with satellite data and related approaches.

Thank you for comment regarding further evaluation of the command areas. We have opted to run the global simulation with all 53 basins using the updated reservoir operations but have left out the command areas. We then evaluated the daily KGE values of this model in conjunction with the previous modelling setups both via CDFs (Figure 1) and through the boxplots (Figure 2). The analysis of the daily KGEs in the CDF show slightly decreased performances for all the stream gages (Figure 1a), while the 6000 stream gages downstream of dams show limited improvements in some regions (Figure 1b). Aggregating these performances into the boxplot in Figure 2 shows that while the variance in KGE values is lower, the average performance without a command area is actually slightly below that of the other command areas. Therefore, we opted to continue our analysis with the Turn250 model.

In addition to including these two figures and the above rational in the Appendix, we have also amended the following lines to include these changes.

- 1. In the abstract on lines 16 18, we include the following: "We also evaluate the sensitivity of our modelling framework to different downstream operating areas (i.e. 0 1100km) and found that there were slight improvements when including downstream demands.
- 2. Lines 528 532 now read as follows: Additionally, the addition of no command areas (Appendix Figure A5a and b) demonstrated very small KGE differences. For the rest of our analysis, we decided not to include the other command areas as a boxplot of the KGEs per command area (i.e. Turn250, Turn600, and Turn1100) depicted little to no variations (Appendix Figure A1)

- and the boxplot of KGEs without the command area (Appendix Figure A1) did not show improvements in KGE.
- 3. Lastly, lines 737 739 now read as follows: We did however observe slight improvements when using command areas compared to without command areas. This suggests that reservoir modelers should use a command area, but can opt to use the command area that makes the most sense to their domain (we opt to use 250).

We have made some minor edits for increased readability and edits to the location of the captions.

Figure 1: Depicts the cumulative distribution function of the daily KGE values of the three main models in our analysis: Baseline (black), vanBeekGeo (grey), Turn250 (pink), and the run with no command areas (orange). Panel a shows the cumulative distribution of the daily KGE for all the GRDC gages, while panel b shows the daily KGE for the 6,000 gages that are downstream of dams.

Figure 2: Shows a boxplot of the data derived operational scheme with the three main command areas: 250, 600 and 1100, in different colors. The Turner scheme with no command area is shown in orange.